# The Lipid- and Polysaccharide-Rich Extracellular Polymeric Substances of *Rhodococcus* Support Biofilm Formation and Protection from Toxic Hydrocarbons

**DOI:** 10.3390/polym17141912

**Published:** 2025-07-10

**Authors:** Anastasiia Krivoruchko, Daria Nurieva, Vadim Luppov, Maria Kuyukina, Irina Ivshina

**Affiliations:** 1Perm Federal Research Center, 13a Lenin Street, 614990 Perm, Russia; daranurieva0@gmail.com (D.N.); kuyukina@iegm.ru (M.K.); ivshina@iegm.ru (I.I.); 2Microbiology and Immunology Department, Perm State National Research University, 15 Bukirev Street, 614990 Perm, Russia; onewayride@vk.com

**Keywords:** extracellular polymeric substances, *Rhodococcus* actinomycetes, biofilms, cell aggregates, bacterial adhesion, tolerance to hydrocarbons, glycosyltransferases

## Abstract

Extracellular polymeric substances (EPS) are multifunctional biopolymers that have significant biotechnological potential. In this study, forty-seven strains of *Rhodococcus* actinomycetes were screened for EPS production and the content of its main components: carbohydrates, lipids, proteins, and nucleic acids. The *Rhodococcus* strains produced lipid-rich EPS (15.6 mg·L^−1^ to 71.7 mg·L^−1^) with carbohydrate concentrations varying from 0.6 mg·L^−1^ to 58.2 mg·L^−1^ and low amounts of proteins and nucleic acids. Biofilms of *R. ruber* IEGM 231 were grown on nitrocellulose filters in the presence of *n*-hexane, *n*-hexadecane, or diesel fuel. The distribution of β-polysaccharides, glycoconjugates, and proteins between cells and the extracellular matrix was examined using fluorescence microscopy. The observed release of β-polysaccharides into the biofilm matrix in the presence of *n*-hexane and diesel fuel was regarded as an adaptation to the assimilation of these toxic hydrocarbons by *Rhodococcus* cells. Atomic force microscopy of the dried EPS film revealed adhesion forces between 1.0 and 20.0 nN, while some sites were highly adhesive (F_a_ ≥ 20.0 nN). EPS biosynthetic genes were identified, with two glycosyltransferases correlating with an increase in carbohydrate production. The production of EPS by *Rhodococcus* cells exhibited strain-specific rather than species-specific patterns, reflecting a high genetic diversity of these bacteria.

## 1. Introduction

Extracellular polymeric substances (EPS) are biopolymers produced by microorganisms into the surrounding media. A large proportion of EPS are considered to be polysaccharides. As a major component of the biofilm matrix, EPS exhibit significant ecological and physiological functions. Furthermore, microbial EPS are multifunctional, biotechnologically valuable compounds. Due to their biodegradability, non-toxicity, water-absorbing capacity, chelating properties, and biological activities, they are used as sources of prebiotics, bioprinting materials, thickeners, emulsifiers, flocculants, stabilizing agents, hydrogels, anti-adhesion preparations, antioxidants, anti-cancer and anti-inflammatory drugs, metal ion binders, plant protectors, and plant growth stimulators. They can be used in medicine, pharmaceuticals, cosmetics, the food industry, remediation, and agriculture [1,2,3,4]. The search for novel producers and the expansion of the range of microbial EPS are deemed key priorities.

Actinomycetes of the genus *Rhodococcus* (domain *Bacteria*, kingdom *Bacillati*, phylum *Actinomycetota*, class *Actinomycetes*, order *Mycobacteriales*, family *Nocardiaceae*, https://lpsn.dsmz.de/genus/rhodococcus, last accessed 16 June 2025) are well-known stress-tolerant biodegraders of a wide range of emergent ecopollutants, mainly hydrocarbons and their derivatives. They are also promising producers of various biotechnologically important metabolites, including amino acids, fatty acids, proteins, carotenoids, biosurfactants, and polysaccharides [5,6]. In comparison with well-characterized polysaccharide EPS, such as alginate produced by *Pseudomonas aeruginosa*, xanthan produced by *Xanthomonas* spp., levan produced by *Paenibacillus polymyxa*, dextran produced by lactic acid bacteria, succinoglycan produced by *Ensifer meliloti* (former *Sinorhizobium meliloti*), and EPS produced by *Bacillus* spp. and various proteobacteria [3,7,8,9], *Rhodococcus* EPS remain poorly characterized.

It is known that EPS produced by *Rhodococcus* bacteria affect their colony morphology, cell-surface hydrophobicity/hydrophilicity, aggregation, flocculation, growth rates, and resistance to toxicants [10,11,12,13]. In particular, EPS in the biofilm matrix enable *Rhodococcus* to resist chlorine-based disinfectants and quinoline, and to dominate in microbial communities in the presence of these compounds [14,15]. Increased EPS production has also been observed in *Rhodococcus* biofilms in the presence of the toxic, hydrophobic dibenzofuran [13]. The protective mechanism of EPS is universal. They trap (absorb) organic xenobiotics and act as a buffer zone between toxicants and cells while increasing adhesion to hydrophobic organic matter [13,14]. However, overproduction of EPS prevents flocculation and cell aggregation, but stimulates cell growth, as demonstrated for *Rhodococcus ruber* TH3 cells of the S- and R-types [12]. Cells of the tetrahydrofuran-degrading strain *R. ruber* YYL with a deleted *gmhD* gene (which encodes D-glycero-D-manno-heptose 1-phosphate guanosyltransferase) do not produce capsule lipopolysaccharides and are able to aggregate with each other [16]. Furthermore, EPS can play a key role in horizontal gene transfer by preventing binding with exogenous DNA. This has been demonstrated in the *R. ruber* YYL Δ*gmhD* mutant. This mutant does not form mushroom-like EPS extrusions, but exhibits high electrotransformation efficiency towards the plasmid pRESQ, with 2.38 × 10^7^ transformed cells per μg DNA. Meanwhile, wild-type cells are surrounded by mushroom-like, translucent EPS and have an electrotransformation ability of less than 10^3^ pRESQ-transformed cells per μg DNA [16].

EPS from *Rhodococcus* show promise as antiflocculants [12], thickeners [17,18], emulsifiers of edible oils [19], antioxidants [19], and cytotoxic agents [18]. They can also be used in bioremediation processes for the biosorption of heavy metals, as well as for use as biosurfactants. These activities have been confirmed for EPS produced by other actinomycetes. EPS produced by marine *Streptomyces* trap Sr^2+^ ions, EPS produced by *Nocardiopsis* trap Cs^+^ ions, and EPS produced by *Arthrobacter* trap Cu^2+^, Pb^2+^, and Cr^6+^ ions. EPS produced by *Streptomyces* exhibit washing and emulsifying activities towards motor oil, various hydrocarbons, and vegetable oils. EPS produced by *Gordonia polyisoprenivorans* emulsify benzene, toluene, and *o*-xylene [20]. EPS from *Rhodococcus opacus* affect the induced mineralization of calcium carbonate and participate in biomineralization. Therefore, they show promise in technologies for conserving stone artwork, the controlled synthesis of inorganic nanophases, crystal engineering of bulk solids, and the assembly of organized composite and ceramic materials [21]. Due to the presence of specific antigens, extracellular polysaccharides isolated from various *Rhodococcus* species (*Rhodococcus coprophilus*, *Rhodococcus equi*, *Rhodococcus erythropolis*, *Rhodococcus rhodnii*, and *Rhodococcus rhodochrous*) bind viral-like particles of the GII.4 and GII.6 genotypes of human norovirus. This seems promising for the development of innovative antiviral strategies to prevent and treat norovirus infections in children [22].

*Rhodococcus* actinomycetes produce various types of EPS. These substances can be fatty acid-containing extracellular polysaccharides [11], acidic polysaccharides [11], or complex mixtures of various polymers [13,21]. The chemical composition of EPS polysaccharides in *Rhodococcus* is well described. These compounds are frequently acidic heteropolysaccharides, with compositions that vary significantly between species and strains. Details of the monomer composition of *Rhodococcus* exopolysaccharides can be found in the literature [11,13,17,18,19,21]. Less is known about the other components of *Rhodococcus* EPS. Non-carbohydrates can prevail or constitute a significant proportion. For instance, lipids account for 5.4% (*w*/*w*) of the fatty acid-containing EPS produced by *R. rhodochrous*. The yield of crude EPS in *R. erythropolis* HX-2 is 6.365 g·L^−1^, with a maximum of 8.957 g·L^−1^, but the yield of exopolysaccharides after multistep purification through the DEAE Cellulose DE-52 and Sepharose CL-6B columns is only 3.736 g·L^−1^. Losses are unavoidable during purification, but it appears that a significant proportion of crude EPS are non-carbohydrates [18]. Saturated fatty acids, such as stearic and palmitic acids, dominate the lipids in *Rhodococcus* EPS [12,23]. A large proportion of proteins are found in EPS produced by *Rhodococcus* sp. p52. These compounds account for 20–33% of total EPS and dominate polysaccharides by 1.5–3.0 times in a fraction of loosely cell-bound EPS [13]. The water-soluble fraction of EPS from *R. opacus* consists of 64.6% polysaccharides and 9.4% proteins, with the remainder being other compounds [21,24]. These proteins may be cytoplasmic molecules that are released into EPS via general secretory pathways and membrane vesicles. Electron transfer (ferredoxin, rubredoxin), toxin–antitoxin, and stress response proteins account for a greater proportion of the proteins present in EPS produced by *Rhodococcus* sp. p52 in the presence of dibenzofuran [13]. Large amounts of proteins are not unique to *Rhodococcus* actinomycetes. For example, the percentage of proteins in the EPS of quinoline-degrading microbial biofilms is fixed at 25–43% [14].

Polysaccharides are typically separated from the other components of EPS and further purified. This is necessary for analyzing their chemical composition and for applications requiring a high level of quality, such as in the fields of (bio)medicine, the food industry, and cosmetology. However, it is crude EPS, rather than their individual components, that determine the true physical-chemical properties of the cell surface. They affect the zeta potential and hydrophobicity of cells, and participate in cell-to-cell contact, as well as contact with substrates, nanoparticles, and surfaces for colonization. They also determine how processes of protection, anti-toxicity, flocculation, and dissolution of contaminants occur [13,14,21]. The hydrolysis of total EPS, including all their components, may affect passive cellular motility, as demonstrated in [13]. The viscosity of EPS (an important characteristic of thickeners and water-holding agents) depends on the combination of the individual components [14].

This study focuses on the crude, loosely cell-bound fraction of EPS produced by various species of the *Rhodococcus* genus, as this is the most cost-efficient fraction for biotechnological purposes. *Rhodococcus* EPS have been studied in terms of their chemical composition, concentrations of major biopolymers, their role in adhesion and formation of biofilms and cell aggregates, and the possible genetic mechanisms of their biosynthesis.

## 2. Materials and Methods

### 2.1. Chemicals

The mineral salts, solvents, Luria–Bertani (LB) broth, D-glucose, crystal violet, Bradford reagent, agar, and Calcofluor White Stain were >97% pure and were purchased from Sigma-Aldrich, Inc. (St. Louis, MO, USA). Phenol was of molecular biology grade and was also purchased from Sigma-Aldrich, Inc. (St. Louis, MO, USA). Concentrated (98%) sulphuric acid was >99.99% pure and was purchased from Chimregionsnab (Ufa, Russia). *n*-Hexane and *n*-hexadecane were >99% pure and were purchased from Cryochrom (St. Petersburg, Russia) and Ekos-1 (Moscow, Russia), respectively. Winter diesel fuel ECTO grade was purchased from Lukoil-Permnefteorgsintez (Perm, Russia). Wheat Germ Agglutinin Alexa Fluor^®^ 350 Conjugate and FilmTracer SYPRO^®^ Ruby Biofilm Matrix Stain were purchased from Thermo Fisher Scientific (Waltham, MA, USA).

### 2.2. Bacterial Strains and Growth Conditions

A total of 47 strains belonging to the following species were used: *Rhodococcus aetherivorans* (2 strains), *Rhodococcus cerastii* (3 strains), *Rhodococcus cercidiphylli* (1 strain), *Rhodococcus corynebacterioides* (1 strain), *R. erythropolis* (8 strains), *Rhodococcus fascians* (1 strain), *Rhodococcus globerulus* (1 strain), *Rhodococcus jostii* (2 strains), *R. opacus* (3 strains), *Rhodococcus pyridinivorans* (2 strains), *Rhodococcus qingshengii* (3 strains), *R. rhodochrous* (8 strains), *R. ruber* (7 strains), *Rhodococcus wtarislaviensis* (1 strain), and *Rhodococcus* sp. (4 strains) (Appendix A). These strains were obtained from the Regional Specialised Collection of Alkanotrophic Microorganisms (acronym IEGM, WDCM number 768, http://www.iegmcol.ru/, last accessed 17 June 2025). The bacteria were grown in 100 mL of LB medium at 160 rpm and 28 °C for 48 h, at an initial cell concentration of 10^6^ colony forming units (CFU) per mL.

### 2.3. Extraction of EPS and Quantitative Analysis of Biopolymers

The fraction of loosely cell-bound EPS was extracted from *Rhodococcus* cells according to [25]. The bacterial cultures were centrifuged at 6000 rpm for 15 min. The supernatant was discarded and the cells were washed with the same volume of phosphate-buffered saline (PBS) at 6000 rpm for 15 min. The PBS contained the following concentrations: NaCl—8.0 g·L^−1^, KCl—2.0 g·L^−1^, Na_2_HPO_4_—1.4 g·L^−1^, KH_2_PO_4_—0.3 g·L^−1^, and pH = 7.4. The cells were then resuspended in 10 mL of PBS and centrifuged at 13,600 rpm for 30 min. The EPS were then precipitated from the supernatant with three volumes of 96% ethanol. The mixture was then left to stand at 4 °C for 24 h before being separated from the solvent at 6000 rpm for 15 min. The EPS were then dried in air and dissolved in 1 mL of Milli-Q^®^ water for subsequent analysis.

The total carbohydrate content of the crude EPS was determined using the phenol-sulphuric acid method [25]. One volume of dissolved EPS was mixed with one volume of 5% phenol and five volumes of concentrated sulphuric acid. The mixture was then incubated in a water bath Elma S10H (Elmasonic, Singen, Germany) at 30 °C for 20 min to produce a red (yellow) color. The color intensity was then measured in quartz cuvettes at 490 nm using a Lambda EZ201 spectrophotometer (Perkin Elmer, Shelton, CO, USA). The protein estimation was performed using the Bradford method. For this, 600 μL of the Bradford reagent was mixed with 60 μL of a sample (EPS dissolved in 1 mL of Milli-Q^®^ water) thoroughly and incubated in the dark at room temperature for 10 min. A color change due to protein binding was observed, and the absorbance was measured at 595 nm. Calibration curves between A_490 nm_ and D-glucose concentration (Appendix A) and between A_595 nm_ and bovine serum albumin concentration (Appendix A) were used to calculate the polysaccharide and protein content, respectively. The nucleic acid content (total DNA and RNA) of the EPS was determined by measuring 1 μL of the EPS samples at 260 nm using a NanoPhotometer N50 (Implen, Munich, Germany), according to the manufacturer’s instructions.

To determine the lipid content of the EPS, a chloroform:methanol 3:1 (*v*/*v*) extraction was conducted [18,26]. For this analysis, dried EPS were used. They were mixed with 4 mL of the solvent mixture, heated at 60 °C for 4 h, and then centrifuged at 5000 rpm for 15 min. The liquid phase was transferred to the pre-weighed tubes and evaporated in a fume cupboard at 45 °C for 2 h, followed by overnight evaporation at room temperature. The dried extracted lipids were allowed to reach constant weight, after which they were weighed using Mettler AC 100 balances (Mettler Toledo, Columbus, OH, USA).

Blank samples (PBS without cells, treated in the same way as samples with EPS) were used as controls in all the quantitative analyses.

### 2.4. Adhesion Tests and Biofilm Growth

Adhesive activity tests were performed in the round-bottom 96-well polystyrene microplates (Medpolymer, St. Petersburg, Russia). A 150 μL suspension of cells in 0.5% NaCl at a concentration of 1·10^8^ CFU·mL^−1^ was transferred to the microplates and incubated in a Titramax 1000 incubator (Heidolph Instruments, Schwabach, Germany) at 600 min^−1^, 28 °C for 24 h. The liquid was then removed and the plates were washed twice with 0.5% NaCl. Next, 150 μL of 1% (*w*/*v*) aqueous crystal violet solution was added. After staining for 20 min at room temperature, the dye was removed and the plates were washed twice with 0.5% NaCl. The crystal violet was then extracted using a 1:4 (*v*/*v*) acetone: ethanol mixture, and the absorbance at 630 nm was measured using a Multiscan Ascent photometer (Thermo Electron Corporation, Vantaa, Finland) [27,28]. A calibration curve between A_630nm_ and CFU was used to quantify the number of adherent cells (Appendix A). Adhesive activities were expressed as the attached cell numbers per unit area (CFU·cm^−2^). Uninoculated 0.5% NaCl was used as an abiotic control.

Biofilms were obtained using nitrocellulose membrane filters with a diameter of 25 mm and pore sizes of 0.22 μm (Millipore, Burlington, MA, USA). Then, 200 μL of a *Rhodococcus* cell suspension with a concentration of 10^6^ CFU·mL^−1^ was dropped onto the filters, which were then placed onto LB agar or mineral agar K plates. Agar K contained the following (http://www.iegmcol.ru/medium/med05.html, last accessed 17 June 2025): KH_2_PO_4_—1.0 g·L^−1^, K_2_HPO_4_—1.0 g·L^−1^, NaCl—1.0 g·L^−1^, KNO_3_—1.0 g·L^−1^, MgSO_4_—0.2 g·L^−1^, CaCl_2_—0.02 g·L^−1^, FeCl_3_—0.001 g·L^−1^, and agar—15.0 g·L^−1^. Paper discs saturated with 200 μL of *n*-hexane, *n*-hexadecane, or diesel fuel were placed under the Petri dish lids and used as co- or growth substrates during cultivation on LB agar or mineral agar K plates, respectively. The plates with inoculated filters were incubated at 28 °C; for 6 days. Petri dishes containing sterile, non-inoculated filters were used as abiotic controls. Inoculated filters placed on LB agar without hydrocarbons were used as a biotic control. The density of the grown biofilms was determined after staining with crystal violet, according to the procedure described above, with modifications. Specifically, the biomass was collected from the membrane filters, resuspended in a 0.5% NaCl solution, and then centrifuged at 6000 rpm for 5 min. The cell pellet was then resuspended in a 1% water solution of crystal violet for staining. The cells were then washed twice with 0.5% NaCl and the crystal violet was extracted from the pellet for subsequent photometric analysis.

### 2.5. Fluorescence Microscopy of Biofilms and Aggregates

The following fluorescent dyes were used to visualize different components in biofilms and cell aggregates: Calcofluor White Stain for β-polysaccharides; Wheat Germ Agglutinin Alexa Fluor^®^ 350 Conjugate for glycoconjugates; and FilmTracer SYPRO^®^ Ruby Biofilm Matrix Stain for proteins. Biofilms were obtained as described above. Their fragments were placed on microscope slides and stained with 20 μL of Calcofluor White in the dark for 1 min. Aggregates formed spontaneously in some flasks. *R. aetherivorans* IEGM 1250 was the only strain that grew in the form of aggregates. A 1 mL of bacterial suspension containing visible aggregates was centrifuged at 4000 rpm for 10 min. The precipitate was suspended in 50 μL of SYPRO Ruby and 50 μL of Wheat Germ Agglutinin, mixed gently and stained in the dark for 20 min. Then, 30 μL of the sample was applied to a microscope slide and 10 μL of Calcofluor White was added. The mixture was then incubated for 1 min in the dark. The cells were then observed using an Axio Imager 2 fluorescence microscope (Carl Zeiss, Oberkochen, Germany) with an excitation wavelength of 380 nm and an emission wavelength of 475 nm.

### 2.6. Atomic Force Microscopy (AFM)

To estimate the adhesion force of EPS, an MFP-3D-BIO™ atomic force microscope (Asylum Research, Santa Barbara, CA, USA) equipped with IgorPro 6.38B01 software (WaveMetrics, Lake Oswego, OR, USA) was used. Dried EPS were dissolved in 100 μL of Milli-Q^®^ water. Then, 10 μL of this 10× concentrated solution was dropped onto a cover glass or a cantilever, after which the drop was left to dry in the air. OMCL-AC24OTS-R3 cantilevers (Olympus, Tokyo, Japan) with a spring constant of 2 N/m and a resonance frequency of 70 kHz, made from silicon, were used. Scanning was performed in contact mode with the following parameters: ForceDist = 2 μm, SpeedScan = 2 Hz and TriggerPoint = 1 V. Height and adhesion force maps measuring 15 × 15 μm, comprising 1024 force curves per sample, were produced.

### 2.7. Bioinformatics Analysis

To identify genes potentially involved in the EPS production, the draft genome sequences of the studied strains, which are available in DDBJ/ENA/GenBank databases (Appendix A), were examined. The genomes were annotated using the NCBI Prokaryotic Genome Annotation Pipeline [29] and RAST 2.0 [30,31]. Specific genes were obtained from the KEGG database (https://www.genome.jp/kegg/pathway.html, last accessed 17 June 2025) and searched in the annotated genome sequences. Paired and multiple nucleotide and amino acid sequence alignments were performed using the NCBI Basic Local Alignment Search Tool (BLAST) (https://blast.ncbi.nlm.nih.gov/Blast.cgi, last accessed 17 June 2025) and ClustalW (https://www.genome.jp/tools-bin/clustalw, last accessed 17 June 2025). Evolutionary analysis and construction of phylogenetic trees were conducted using MEGA12 (https://www.megasoftware.net/, last accessed 17 June 2025) [32]. The neighbor-joining method was used to construct the trees. Branch lengths reflecting evolutionary distances (similarities) were calculated using the maximum likelihood method. A search for biosynthetic gene clusters (BGCs) was performed using antiSMASH 7.0 [33].

### 2.8. Statistics

All experiments were performed in three independent trials, with 3–8 replicates. Statistica version 13.5.0.17 (TIBCO Software Inc., Palo Alto, CA, USA) and Julius (https://julius.ai, last accessed 4 June 2025) were used to calculate basic statistics, compare data, perform correlation analysis, and create heatmaps. IgorPro 6.38B01 and Claude Sonnet 4 AI model (Anthropic, San Francisco, CA, USA) were used to treat and analyze the AFM data.

## 3. Results

### 3.1. The Chemical Composition of EPS Produced by Rhodococcus spp.

The concentrations of the major biological polymers (polysaccharides, lipids, proteins, and nucleic acids) in loose EPS produced by *Rhodococcus* bacteria were estimated. A phenol-sulphuric acid reaction was used to detect polysaccharides. However, this reaction is not strictly specific to these compounds, determining the total content of carbohydrate residues, which could originate from polysaccharides, glycoconjugates (e.g., glycolipids and glycoproteins), lipopolysaccharides, nucleic acids, monosaccharides, disaccharides, and oligosaccharides present in the EPS. Furthermore, the term ‘EPS carbohydrates’ was used, essentially assuming that the carbohydrates originated predominantly from polysaccharides. As seen from Table 1 and Figure 1a, the studied *Rhodococcus* strains produced low amounts of EPS carbohydrates, ranging from 0.6 ± 0.2 mg·L^−1^ (*R. aetherivorans* IEGM 1250) to 58.2 ± 24.0 mg·L^−1^ (*R. ruber* IEGM 231). The median EPS carbohydrate production was as low as 8.9 mg·L^−1^.

The production of the EPS carbohydrates was independent of the taxonomic classification of the strains, as this parameter varied significantly between representatives of the same species (Figure 1a). Accurate correlation analysis was performed on species with ≥3 strains, including *R. erythropolis/R. qingshengii*, *R. cerastii*, *R. rhodochrous* and *R. ruber* (Figure 1b). The Kruskal–Wallis test revealed no statistically significant differences between species (*p* = 0.61). The correlation coefficient for the EPS carbohydrate production and *Rhodococcus* species was as low as R_Spearman_ = 0.05 at *p* = 0.82 (Figure 1b).

A subset of strains representatives of various *Rhodococcus* species was used to analyze other common biopolymers present in EPS, such as lipids, proteins, and nucleic acids (Table 2). High concentrations of lipids were detected, with production of these compounds varying from 15.6 ± 1.6 mg·L^−1^ (*Rhodococcus* sp. IEGM 1401) to 71.7 ± 7.2 mg·L^−1^ (*R. erythropolis* IEGM 1415), with a median value of 32.0 mg·L^−1^. This was four times higher than the median production of EPS carbohydrates (Table 2, Figure 2). Concentrations of proteins and nucleic acids in *Rhodococcus* EPS were negligible, consisting of no more than 1.131 ± 0.091 mg·L^−1^ for proteins and 2.735 ± 1.276 mg·L^−1^ for nucleic acids (Table 2). The highest values of total proteins and nucleic acids were found only in the EPS produced by *R. rhodochrous* IEGM 107. Other strains produced less than 0.5 mg·L^−1^ of these compounds (Table 2, Figure 2).

The proportions of EPS biopolymers in individual *Rhodococcus* strains are shown in Figure 2. In one strain (*R. erythropolis* IEGM 1399), carbohydrates predominated over lipids by 56%. In three other strains—*R. globerulus* IEGM 1203, *R. wratislaviensis* IEGM 1171 and *R. opacus* IEGM 249—lipids did not significantly prevail over carbohydrates at 57%, 70%, and 48%, respectively. In seven strains, concentrations of EPS lipids exceeded those of EPS carbohydrates by 2–6 times. The EPS of other three strains (*R. erythropolis* IEGM 1415, *R. pyridinivoranss* IEGM 66 and *R. ceastii* IEGM 1327) consisted almost entirely of lipids. They predominated over carbohydrates by 15–33 times (Figure 2). Again, no relationship was observed between species and the composition of the studied EPS preparations.

### 3.2. EPS in Rhodococcus Biofilms and Cell Aggregates

The distribution of EPS between cells and the extracellular matrix was studied in *Rhodococcus* biofilms and cell aggregates using fluorescence microscopy. *R. ruber* IEGM 231 was selected as a model strain for biofilm formation. This strain is a known biodegrader of a wide range of hydrocarbons (http://www.iegmcol.ru/strains/rhodoc/ruber/r_ruber231.html, last accessed 17 June 2025) [34,35] and produces the highest amounts of EPS carbohydrates (Table 1). *R. ruber* IEGM 231 formed biofilms on all nitrocellulose membrane filters. As an oligotrophic microorganism [5], the IEGM 231 formed weak biofilms ((0.9 ± 0.1)·10^9^ CFU·cm^−2^) even in minimal media, such as mineral agar K without the supplementation of an external carbon source (Figure 3). The density of *R. ruber* IEGM 231 biofilms was affected by hydrocarbons. In particular, *n*-hexane, *n*-hexadecane, and diesel fuel resulted in a 2–8-fold increase in biofilm growth on mineral agar. This effect was observed in biofilms grown on mineral agar only, and was most notable for toxic hydrocarbons (e.g., *n*-hexane and diesel fuel), and least notable for *n*-hexadecane. No statistically significant differences were revealed between the biofilm densities of the biotic control and the hydrocarbon exposure on the LB agar (Figure 3).

Staining of *R. ruber* IEGM 231 biofilms with Calcofluor White revealed that the β-polysaccharides were bound to and surrounded the cells, as well as constituting the extracellular matrix (Figure 4). The intensity of the fluorescence reflected the quantity of β-polysaccharides present on the cells and in the matrix. When exposed to *n*-hexadecane, *R. ruber* IEGM 231 cells exhibited a bright glow, indicating the presence of large quantities of β-polysaccharides bound to and surrounding the cells. The amounts of β-polysaccharides in the extracellular matrix were low. There was hardly any fluorescence, or only weak fluorescence, in the space between cells in the biofilms grown in the presence of *n*-hexadecane (Figure 4b,e).

When exposed to toxic hydrocarbons (e.g., *n*-hexane and diesel fuel), the *R. ruber* IEGM 231 cells appeared less bright, and the extracellular matrix appeared evidently white-blue (Figure 4c,f). Apparently, there were fewer cell-bound β-polysaccharides and more matrix-associated ones, indicating the release of β-polysaccharides into the extracellular space. In the presence of diesel fuel, as well as on LB agar without hydrocarbons, the cells were surrounded by loose, wide, fuzzy β-polysaccharide capsules. However, this effect was more pronounced at the diesel fuel exposure (Figure 4a,c). The formation of capsules was not observed in biofilms grown in the presence of individual *n*-alkanes or on mineral agar K without externally added growth substrates (Figure 4b,d–f). Cells in weak oligotrophic biofilms formed on agar K produced small amounts of the cell-bound β-polysaccharides and did not produce matrix β-polysaccharides, as evidenced by weak fluorescence (Figure 4d). This confirmed that no mature and functional rhodococcal biofilms were formed in this case. It is important to note that the effects of hydrocarbons on the distribution of β-polysaccharides between cells and the extracellular matrix were similar for biofilms grown on both LB agar and mineral agar K. However, their influence on the biofilm density was not registered on LB agar (Figure 3).

The distribution of EPS was also examined in rhodococcal cell aggregates. One strain, *R. aetherivorans* IEGM 1250, grew in the form of aggregates. In other species, this process was more random and spontaneous, with typical growth occurring in the form of homogeneous suspensions. Aggregation was only rarely observed, specifically in *R. erythropolis* IEGM 1020 and *R. jostii* IEGM 68 (Figure 5). All three strains produced low amounts of EPS carbohydrates (0.6–6.7 mg·L^−1^) (Table 1). Furthermore, *R. jostii* IEGM 68 produced low amounts of EPS lipids (17.3 ± 9.1 mg·L^−1^) (Table 2). Aggregation apparently did not affect the total EPS production, as no relationship was found between concentrations of EPS biopolymers and the formation of cell aggregates in particular flasks (i.e., replicates).

Aggregates of *Rhodococcus* cells were immersed in a matrix consisting of a complex mixture of β-polysaccharides and glycoconjugates, both of which produced a similar color when examined using fluorescence imaging. These compounds were found to be associated with the cells and the extracellular matrix in a manner similar to that observed in biofilms (Figure 5). *R. aetherivorans* IEGM 1250 cells formed dense aggregates with little free matrix (Figure 5a,b), whereas *R. erythropolis* IEGM 1020 cells (Figure 5c) and *R. jostii* IEGM 68 cells (Figure 5d) formed relatively loose aggregates with lower cell densities and smaller amounts of cell-bound EPS and more extracellular matrix. The synthesis of cell-associated EPS differed between individual cells. Some cells were brighter and produced more EPS. Other cells were paler and produced fewer EPS. Some cells were unevenly stained, representing uneven distribution of EPS on the cell surface (Figure 5).

According to the multi-component staining data, the proteins were not part of the extracellular matrix of the *Rhodococcus* cell aggregates. Red, orange-red, and pink fluorescence associated with proteins was detected on the surface of bacterial cells rather than in the matrix (Figure 5). The proportion of cells synthesizing proteins differed between strains. Many red-orange cells were found in *R. erythropolis* IEGM 1020 aggregates (Figure 5c). *R. aetherivorans* IEGM 1250 aggregates predominantly consisted of cells with white/blue fluorescence that were actively synthesizing carbohydrate EPS. However, a proportion of cells producing surface proteins were also observed (Figure 5a,b). In the *R. jostii* IEGM 68 aggregates, protein-rich cells were localized at the poles of the aggregates (Figure 5d).

### 3.3. The Involvement of EPS in the Adhesion of Rhodococcus spp. To Solid Surfaces

The results of adhesion tests for *Rhodococcus* cells to polystyrene are shown in Appendix A. No statistically significant correlation (R_Spearman_ ≤ 0.41, *p* > 0.05) was revealed between the adhesive activity of rhodococci towards polystyrene and the amounts of carbohydrates and lipids in EPS. Two strains, *R. opacus* IEGM 249 and *R. rhodochrous* IEGM 1162, exhibited enhanced adhesive abilities. Some 63–64% of cells of these strains adhered to polystyrene, equivalent to 4.176–4.248·10^7^ attached cells per cm^2^ (Appendix A). These strains produced low amounts of EPS lipids (19.0 ± 7.9 mg·L^−1^) and median amounts of EPS carbohydrates (9.2–12.8 mg·L^−1^) (Table 1 and Table 2). Eight strains were unable to adhere to polystyrene (Appendix A). Their synthesizing activities varied widely from 0.6 ± 0.2 mg·L^−1^ to 23.9 ± 13.5 mg·L^−1^ for the EPS carbohydrates and from 17.3 ± 9.1 mg·L^−1^ to 71.7 ± 7.2 mg·L^−1^ for the EPS lipids (Table 1 and Table 2). The adhesive activities of the other *Rhodococcus* strains were between (0.153 ± 0.029)·10^7^ CFU·cm^−2^ and (1.730 ± 0.380)·10^7^ CFU·cm^−2^, corresponding to 2% and 26% of the attached cells, respectively (Appendix A). Their EPS production abilities also varied significantly (Table 1 and Table 2).

The adhesive potential of the *Rhodococcus* EPS was additionally estimated by measuring the adhesion force F_a_ (nN) of the EPS produced by *R. ruber* IEGM 231. These EPS appeared to be inhomogeneous under an atomic force microscope. This could be seen on topographic maps of the EPS drop surface as a combination of dark and light spots, which corresponded to low- and high-relief areas, respectively (Figure 6c,d). This could be related to different densities and the formation of cords or strands consisting of more condensed EPS material [36]. The revealed heterogeneity was not associated with an arrangement of adhesion forces. Adhesive (light) and non-adhesive (dark) spots were sporadically located across the surface of EPS drops (Figure 6g). The median F_a_ for the EPS produced by *R. ruber* IEGM 231 was 7.0 nN, which was 2.8 nN (29%) lower than the median F_a_ of unmodified cover glass (Figure 6e–h). This could be due to electrostatic repulsion between the silicon cantilever and the charged EPS saturated with -OH groups. Most sites of EPS produced by *R. ruber* IEGM 231 exhibited adhesion forces between 1.0 and 20.0 nN. However, some rare sites were highly adhesive, with F_a_ ≥ 20.0 nN (Figure 6h). Unmodified glass did not exhibit such adhesive spots, with its highest F_a_ values being around 13.2 nN (Figure 6e,f).

The adhesion force of the AFM cantilever modified with EPS towards the unmodified cover glass was 8.0 nN, which was 1.0 nN higher than that of the unmodified cantilever (Figure 7c). However, a bimodal distribution of adhesion forces was revealed. Some 80% of the measurements were >5.5 nN, with a median value of 8.2 nN; and 15% of the measurements were ≤5.5 nN, with a median value of 3.2 nN (Figure 7d). The roughness of the cover glass when scanned with the EPS-modified cantilever was the same as when scanned with the unmodified cantilever (Figure 7a,b). This provided evidence that the scanning was correct. Additional topographic and adhesion force maps can be found in Appendix A.

### 3.4. Genes Involved in EPS Biosynthesis in Rhodococcus spp.

Genes likely involved in the biosynthesis of EPS polysaccharides in *Rhodococcus* spp. were analyzed. These genes coded for glycosyltransferases (gtfs), capsular polysaccharide biosynthesis proteins, undecaprenyl-phosphate galactose phosphotransferases, homoserine O-acetyltransferases, N-acetylglucosaminyl-diphospho-decaprenol L-rhamnosyltransferases, polysaccharide biosynthesis proteins, and acyl-CoA:1-acyl-sn-glycerol-3-phosphate acyltransferases. The latter two types of enzymes were rare, being found only in two genomes: *Rhodococcus* sp. IEGM 1408 and *R. rhodochrous* IEGM 1360, respectively. The genes coding for glycosyltransferases were the most abundant and were found in all analyzed genomes (Figure 8, Appendix A). The studied strains mainly harbored more than one glycosyltransferase gene. The largest number of glycosyltransferase genes detected was six, in three strains: *R. cerastii* IEGM 1243, *R. corynebacterioides* IEGM 1202 and *Rhodococcus* sp. IEGM 1408. The other three strains (*R. erythropolis* IEGM 1321, *R. ruber* IEGM 1391, and *Rhodococcus* sp. IEGM 1414) harbored five glycosyltransferases. Nine, nine, and eight strains had four, three, and two different glycosyltransferases, respectively. Only four strains (*R. rhodochrous* IEGM 107, IEGM 1161, IEGM 1362 and *Rhodococcus* sp. IEGM 1409) harbored one glycosyltransferase (Figure 8).

As can be seen from the phylogenetic tree, the glycosyltransferases of *Rhodococcus* were diverse (Figure 9). Thirteen distinct clades could be identified based on bootstrap support. This parameter varied from 38% up to 99%. Most of these clades had a bootstrap support value of 94–99%. Four clades had lower support (38–58%), but these were further divided into low-rank peripheral groups with strong (98–99%) confidence. Therefore, glycosyltransferase-encoding genes within these confident clades were homologous and highly similar, sharing a common origin. However, they had no phylogenetic relationship with glycosyltransferases from other clades. Their divergence appears to have occurred long ago and could not be confirmed using sequence alignments. This can be seen from the very low (≤13%) bootstrap support for nodes on deep branches (Figure 9).

The studied *Rhodococcus* strains harbored glycosyltransferase enzymes from different clades. Only some strains harbored two homologous enzymes from one clade. These strains were *R. cerastii* IEGM 1243 (gtfs 2 and 3), *R. corynebacterioides* IEGM 1202 (gtfs 1 and 2), *R. erythropolis* IEGM 1321 (gtfs 3 and 4), *R. fascians* IEGM 1233 (gtfs 2 and 4), *R. globerulus* IEGM 1203 (gtfs 1 and 2), *R. ruber* IEGM 560 (gtfs 1 and 2), and *Rhodococcus* sp. IEGM 1414 (gtfs 2 and 5). Their enzymes were not identical copies, except for gtfs 1 and 2 in *R. corynebacterioides* IEGM 1202 and gtfs 1 and 2 in *R. ruber* IEGM 560, which were completely identical to each other (Appendix A). Divergence was observed among glycosyltransferases within clades, as evidenced by the low (18–84%) support for some peripheral groups (Figure 9) and the varying branch lengths on the amino acid similarity tree (Appendix A). The nucleotide sequence tree reflected the evolutionary and historical divergence of *Rhodococcus* species more accurately, although this was not necessarily reflected in divergence of glycosyltransferase functions. The amino acid sequences of glycosyltransferases were less diverse, and gtfs from various clades (particularly clades 8, 11 and 12) formed one large group with high similarity on the amino acid tree. This is the largest (32 gtfs) upper group in Appendix A. Differences in branch length within this group were no greater than 0.1, and at least one copy of the enzyme from this group was found in almost all of the strains studied.

Some patterns were observed among the species. Strains of *R. rhodochrous* had fewer glycosyltransferases (1–2), but harbored five undecaprenyl-phosphate galactose phosphotransferases. *R. cerastii* had the greatest number of glycosyltransferases (4–6) (Figure 8 and Figure 9). The glycosyltransferases of *R. erythropolis* and *R. qingshengii* predominated in two large clades, 8 and 13. In other clades, these species were absent or present in lower numbers (Figure 9). The two *R. opacus* strains analyzed were highly similar and had identical sets of four different glycosyltransferases (Figure 9 and Appendix A). *R. corynebacterioides* IEGM 1202 and *Rhodococcus* sp. IEGM 1408 had unique combinations of glycosyltransferases. Their amino acid sequences were on separate branches and differed significantly from other gtfs (Appendix A).

A statistically significant correlation (R = 0.39, *p* < 0.05) was found between concentrations of EPS carbohydrates and the presence of the glycosyltransferase from clade 6 (C on the protein tree) (Figure 10). This glycosyltransferase was found in five strains, including two *R. opacus* strains (IEGM 249 and IEGM 2226) and three *R. ruber* strains (IEGM 231, IEGM 1121 and IEGM 1263). These strains produced EPS carbohydrates at a level of ≥10.5 mg·L^−1^, which was higher than the median production level (Table 1). The most efficient producer of EPS carbohydrates, *R. ruber* IEGM 231, was among these five strains. Additionally, the IEGM 231 strain and the *R. opacus* strains harbored another glycosyltransferase from a clade 4 on the gene tree (Figure 9) and a distinct clade D on the protein tree (Appendix A). There was a strong correlation (R = 0.47, *p* < 0.05) between concentrations of EPS carbohydrates and the presence of this enzyme (Figure 10). Both glycosyltransferases were phylogenetically related and located in neighboring groups. The evolutionary distances between these glycosyltransferases in *R. opacus* and *R. ruber* were as low as 0.2 (Figure 9 and Appendix A). The presence of other genes, as well as the number of glycosyltransferases, did not influence EPS carbohydrate production by *Rhodococcus* cells. This was evidenced by low correlation coefficients and *p*-values ≥ 0.05 (Figure 10).

No BGCs specific to exopolysaccharide synthesis were found in the *Rhodococcus* genomes (Table 3). Only a cluster containing genes 75% similar to those for ectoine biosynthesis was found in the *R. rhodochrous* IEGM 107 genome. Clusters that are not polysaccharide-related or BGCs with very low (≤27%) similarity were found in genomes of other strains. The annotated BGCs were related to the biosynthesis of osmoprotectants, siderophores, terpenes, antibiotics, alkaloids, and cell wall components (particularly glycopeptidolipids). These BGCs did not harbor specific genes for the biosynthesis of exopolysaccharides, but rather contained genes for the glycosylation of the major product or its intermediates in the series of biochemical reactions connected with these BGCs (Table 3). Further details about the BGCs identified are summarized in Appendix A.

## 4. Discussion

The ability of 43 strains belonging to 14 known *Rhodococcus* species, as well as four *Rhodococcus* sp. isolates, to produce EPS, and the chemical composition of these EPS, were studied. In particular, the characteristics of crude, loosely cell-bound *Rhodococcus* EPS were examined. This fraction was selected for the analysis as it could be obtained routinely using only centrifugation steps. Our primary focus was on the EPS carbohydrates in this fraction. We assumed that the dominant compounds were exopolysaccharides. The *Rhodococcus* strains used produced low amounts of EPS carbohydrates. The median production of these components was 8.9 mg·L^−1^, with the highest production being 58.2 mg·L^−1^ by the *R. ruber* IEGM 231 cells. This production level was similar to that of exopolysaccharides produced by mono- and dual-species biofilms of sludge bacteria and *Rhodococcus* sp. BH4, which ranged from 8 to 390 mg·L^−1^ [37]. However, other studies have shown that *Rhodococcus* strains can synthesize 10–1000 times more exopolysaccharides, with concentrations ranging from 3.7 to 9.0 g·L^−1^ [18,19]. The low biosynthetic abilities of IEGM strains may be due to underestimated carbohydrate content in the obtained EPS or non-optimal conditions for the EPS production. In studies [18,19], *Rhodococcus* cells were heated at 90 °C for 15–30 min prior to EPS separation. This preliminary procedure released additional EPS that were more tightly bound to the cell envelope. Heating at 60 °C for 30 min was also used to separate EPS from *Rhodococcus* sp. BH4 biofilms [37,38], which may account for the higher amounts of EPS observed in those studies. Fluorescence microscopy photographs of *R. ruber* IEGM 231 biofilms show that polysaccharides are concentrated on the surface of *Rhodococcus* cells rather than in the matrix (Figure 4). It is likely that polysaccharides constituted tightly bound EPS, as well as cell capsules, rather than the loose fraction. The bright coloring of *Rhodococcus* cells in biofilms after staining with Calcolfluor White, which is specific to β-polysaccharides, provides evidence of the widespread presence of these compounds among the *Rhodococcus* EPS carbohydrates.

The growth conditions in this study were optimal for *Rhodococcus* cells. They included a rich growth medium (LB), an optimal growth temperature of 28 °C, and good aeration on an orbital shaker. EPS typically exhibit protective functions and facilitate substrate assimilation [10,13,14,39]. This study confirmed these functions in dense *R. ruber* IEGM 231 biofilms formed in the presence of toxic *n*-hexane and diesel fuel. The release of β-polysaccharides into the extracellular matrix was stimulated in the dense biofilms (Figure 4). Exopolysaccharides were probably not essential for *Rhodococcus* cultures in LB but were necessary to enhance protection of *Rhodococcus* cells from toxic hydrocarbons. The optimization of growth conditions to enhance EPS production by microbial cells is a separate step. It includes selecting multiple factors, such as pH, temperature, salt concentrations, aeration regimes, growth substrates, and nitrogen sources, as well as their concentrations, supported by mathematical modeling [18,40]. In this study, we compared various *Rhodococcus* strains, with optimization planned as the next step.

The most surprising result was the detection of large amounts of lipids in the EPS. It was already known that lipids were present in large quantities on the surface of the rhodococcal cell envelope [11,12,18,23]. In this study, however, they prevailed over EPS carbohydrates, and this disproportion was incredible: 15–33 times for certain strains. Lipids were not visualized in this study using fluorescent dyes. The *R. ruber* IEGM 231 biofilms were stained with Calcofluor White only, which is specific to β-polysaccharides, to highlight the distribution of these primary compounds of our interest between cells and the biofilm matrix. However, another study [41] showed that the extracellular matrix of *Rhodococcus* biofilms was saturated with lipids and had an intense red coloring after staining with Nile red. This confirmed our findings on the predominance of lipids in loose EPS. The mixture of solvents (chloroform:methanol, 3:1) used for lipid extraction could theoretically result in the extraction not only acylglycerols and free fatty acids, but also complex lipid components, particularly lipopolysaccharides, glycolipids, lipoproteins, and glycopeptidolipids. However, this assumption was not supported by data on the low concentrations of EPS carbohydrates and proteins, which could act as putative ligands for fatty acid residues.

The amounts of proteins in the EPS produced by the studied *Rhodococcus* strains were negligible and were below 1 mg·L^−1^ for most strains (Table 2, Figure 2). However, their notable presence in loose *Rhodococcus* EPS has been reported in other studies [13,21,24]. In study [13], the accumulation of proteins in the EPS produced by *Rhodococcus* sp. p52 was a response to the presence of dibenzofuran. However, rhodococci grew under optimal conditions in this study. Studies [21,24] analyzed concentrated, purified EPS from *R. opacus*, which could result in the overestimation of the true protein concentrations. Fluorescence microscopy images of *Rhodococcus* cell aggregates showed that proteins were localized on the surface of cells and were not present in the extracellular matrix (Figure 5). This was consistent with the results of multi-component staining of *R. rhodochrous* IEGM 1363 biofilms previously conducted. In those biofilms, proteins surrounded cells and were not present in the matrix [41].

The presence of large amounts of lipids and small amounts of proteins in loose EPS produced by the IEGM strains makes these EPS promising emulsifiers for removing hydrocarbons, and they are potentially suitable as food or cosmetic additives. Lipids can enhance the emollient properties of EPS preparations, while the minor presence of proteins reduces the risk of allergies. However, purification and detailed chemical analysis are certainly required, and will be carried out in subsequent studies.

No dependence of EPS production or adhesive activity on species was observed among the studied *Rhodococcus* strains. The strain-specific nature of functional traits is typical of *Rhodococcus* actinomycetes. This has been demonstrated with regard to their adhesive activities towards solid hydrocarbons, their abilities to degrade aromatic compounds, and their potential to accumulate heavy metals [34,42,43]. The reasons could be genetic versatility and redundancy, as well as the high genetic heterogeneity of *Rhodococcus* bacteria within species [44,45]. This has been clearly demonstrated for genes encoding enzymes of the exopolysaccharide biosynthesis. Although species patterns were found for these genes, they tended to be disarranged. Notably, *R. rhodochrous* strains had five undecaprenyl-phosphate galactose phosphotransferases, one glycosyltranferase from the clade 12, 1–2 capsular polysaccharide biosynthesis proteins, and 1–2 homoserine O-acetyl transferases. However, two *R. rhodochrous* strains lacked undecaprenyl-phosphate galactose phosphotransferase genes, and one *R. rhodochrous* strain lacked homoserine O-acetyl transferase genes (Figure 8). Gene combinations in strains of other species were even more diverse. Notably, glycosyltransferases specific to *R. erythropolis* and *R. qingshengii* were found in clades 8 and 13. The gtf 8 was detected in all strains. The gtf 13 was absent in one strain and characterized by a high diversity in amino acid sequences (it was found in clades G, E, and F on the protein tree). Additionally, various combinations of other gtfs were found in *R. erythropolis* and *R. qingshengii* strains (Figure 8, Figure 9 and Appendix A).

The genetic diversity of *Rhodococcus* bacteria could be related to horizontal gene transfer (HGT) events and the possible localization of genes on megaplasmids [46,47]. *Rhodococcus* strains within one species exhibit significant genomic diversity. Large fragments of DNA can be lost or transferred, and these events are specific to individual strains, rather than species. This has been demonstrated in the case of soluble di-iron monooxygenases (SDIMOs) responsible for propane oxidation in *R. ruber*. It is known that a set of genes (operon) codes for one molecular complex of SDIMO. *R. ruber* has two SDIMO sets, one of which is linked with mobile elements and lost in some strains [6]. Intrinsic gene diversity, in combination with the existence of alleles and differences in the gene expression levels, can lead to significant variations in degrading and biosynthetic activities between strains within one *Rhodococcus* species.

The gene combinations did not influence the production of EPS carbohydrates. Only the presence of two glycosyltransferases specific to *R. opacus* and *R. ruber* enhanced the biosynthesis of EPS carbohydrates (Figure 10). These gtfs could be used in the future to identify and predict promising EPS producers. As previously mentioned, this could be related to differences in gene expression. Another possible reason is the relationship between the genes and amounts of tightly bound EPS, which were not assessed in this study. The combinations of genes may be more important for the chemical composition of the exopolysaccharides produced by *Rhodococcus* bacteria. Genetic variability could explain differences in composition of *Rhodococcus* EPS reported in the literature. In particular, the exopolysaccharides produced by *R. erythropolis* HX-2 are composed of glucose, mannose, galactose, glucuronic acid, and fucose, with mass ratios of 27%, 27%, 25%, 16%, and 5%, respectively [18]. The exopolysaccharide mucoidan produced by another strain of the same species, *R. erythropolis* PR4, was different, consisting of a pentasaccharide repeating unit of two β-D-glucose, one β-D-N-acetylglucosamine, one α-D-glucuronic acid, and one α-L-fucose residue [11]. This highlights the importance of in-depth studies of the genetic mechanisms responsible for EPS biosynthesis in *Rhodococcus* in order to select strains that produce specific EPS. It has also been shown that genetic modifications and the molecular engineering of EPS biosynthesis could present a challenge in *Rhodococcus* cells. No genes involved in (exo)polysaccharide biosynthesis were found to be connected with BGCs in the IEGM strains (Table 3). These genes probably participate in multiple biosynthetic pathways and are responsible not only for the biosynthesis of polysaccharide chains, but also for the glycosylation of various monomeric and polymeric metabolites, e.g., antibiotics, terpenes, amino acids, nucleotides, lipids, proteins, etc. Their modification or regulation of their expression levels can affect cell physiology in unpredictable ways.

The lack of association between putative polysaccharide biosynthetic genes and BGCs suggests that other regulatory mechanisms may govern exopolysaccharide biosynthesis in *Rhodococcus*. Analyzing the surrounding area of the studied biosynthetic genes revealed the presence of a few genes coded for transcriptional regulators. These belong to the AraC, PhoU, XRE (xenobiotic response element), and OmpR families. The AraC (https://www.ncbi.nlm.nih.gov/Structure/sparcle/archview.html?archid=13263893, last accessed 8 July 2025), XRE (https://www.ebi.ac.uk/interpro/entry/cdd/CD02209/, last accessed 8 July 2025), and OmpR (https://www.uniprot.org/uniprotkb/P0AA18/entry, last accessed 8 July 2025) participate in the response of bacterial cells to stress, particularly the presence of xenobiotics in the case of XRE and osmotic stress in the case of OmpR. AraC is responsible for the metabolism of L-arabinose and the control of the degradation of several sugars. PhoU regulates the metabolism of phosphorus [48], which, in the form of phosphates, is required for the transfer of glycosyl residues. We hypothesize that these transcriptional regulators can control the expression of (exo)polysaccharide biosynthetic genes in *Rhodococcus*. Additionally, their proximity to the studied genes, which mainly encode glycosyltransferases, capsular polysaccharide biosynthesis proteins, and undecaprenyl-phosphate galactose phosphotransferases, suggest a role for EPS in protecting *Rhodococcus* cells from toxic hydrocarbons. For example, OmpR is part of a two-component sensor regulatory system. This system is responsible for the adaptation of bacteria to osmotic stress and reacts to the changes in the cell microenvironment. It can sense the disruptive effects on the cell wall and cytoplasmic membrane caused by toxicants such as *n*-hexane or diesel fuel. Further, it can impact the expression of genes involved in glycosylation and the transfer of glycosylated substances. This can result in the release of EPS components into the extracellular matrix, similar to the release of β-polysaccharides in the *R. ruber* IEGM 231 biofilms.

Other genes in proximity to the putative genes coded for (exo)polysaccharide biosynthesis in *Rhodococcus* are connected with various processes. These genes encode transporters and enzymes that are likely to be involved in oxidation/degradation (oxidoreductases), resistance (e.g., chloramphenicol acetyltransferase), and the metabolism of lipids, terpens, steroids, carbohydrates, amino acids, and nucleotides (various transferases, synthetases, dehydrogenases, dehydratases, etc.). Many genes are annotated as hypothetical proteins. This supports the hypothesis that EPS biosynthesis genes in *Rhodococcus* are involved in diverse metabolic processes and are not specific to the biosynthesis of (exo)polysaccharides. Genes coded for capsular polysaccharide biosynthesis proteins are always located next to any glycosyltransferase. Genes encoding undecaprenyl-phosphate galactose phosphotransferases are located in one sector, with one or a few glycosyltransferases. These genes may be in direct proximity or separated by several other genes. We also hypothesize that these gtfs transfer galactose. The revealed gene distribution provides evidence for their coupling and involvement in one chain of metabolic reactions. Mobile elements were not detected in genome segments with polysaccharide biosynthesis genes. It appears that the diversity of rhodococcal gtfs is the result of evolutionary changes, e.g., divergence and convergence.

Controversial results were obtained regarding the role of EPS in *Rhodococcus* cell adhesion and biofilm growth. On the one hand, EPS were found to be important in protecting rhodococci from toxic hydrocarbons. This was evident from the increased biofilm density and more pronounced extracellular matrix, as well as from the formation of polysaccharide-wrapped cells, and the release of exo-β-polysaccharides into the biofilm matrix when exposed to *n*-hexane and diesel fuel (Figure 3 and Figure 4). These reactions seem to be protective rather than facilitating hydrocarbon utilization. No differences were observed in biofilm density on LB agar, either with or without hydrocarbons. However, β-polysaccharides were released into the matrix of biofilms grown on LB in the presence of toxic *n*-hexane and diesel fuel. β-Polysaccharides were apparently required for cell protection, but they did not induce hydrocarbon assimilation. The release of β-polysaccharides did not occur in the presence of non-toxic *n*-hexadecane in either medium, as no enhancement of matrix buffer functions was apparently required. However, some involvement of EPS in the utilization of hydrocarbons cannot be ruled out. A synergetic effect was probably observed on agar K in the presence of *n*-hexane and diesel fuel, on which rhodococcal cells formed the 3–8-fold denser biofilms compared to the control and the 2–5-fold denser biofilms compared to *n*-hexadecane. The resistant cells were apparently able to effectively assimilate *n*-hexane and diesel fuel. This may be related to the accumulation of hydrocarbon molecules in the matrix and the optimized rate at which they were transported to the cells.

On the other hand, the adhesion of *Rhodococcus* cells to polystyrene did not depend on the production of EPS carbohydrates; the adhesion forces of *Rhodococcus* EPS towards glass and the AFM cantilever were weaker than those in the control measurements (without EPS), and cell aggregation was detected in strains with a low production of EPS carbohydrates. The latter was not surprising. In study [12], R-type cells of *R. ruber* TH3 produced low amounts of EPS and formed aggregates, whereas S-type cells produced large amounts of EPS and remained suspended. The *R. ruber* YYL cells, which had lost their ability to synthesize a capsule, aggregated easily. This can be explained by the presence of many negatively charged -OH and -COOH groups in the composition of the EPS polysaccharides, resulting in cell retraction [16]. Apparently, EPS were not key factors in the adhesion of *Rhodococcus* cells to solid surfaces in the first step of the biofilm formation. However, they (particularly EPS polysaccharides) were important for biofilm growth and tolerance to toxicants. An increase in polysaccharide amounts in the extracellular matrix has been observed in microbial biofilms in pipes treated with chlorine [15], as well as in quinoline-degrading biofilms in a membrane aerated biofilm reactor [14]. Cultivating *Rhodococcus* bacteria in the presence of pollutants can be an effective and economical way to increase the production of loose EPS by these microorganisms.

The EPS produced by *Rhodococcus* cells contains a high proportion of lipids. These lipids can play a significant role in the assimilation of hydrocarbons by rhodococcal cells. Lipids attract hydrocarbons, which are trapped and dissolved in the lipid-rich matrix. This probably facilitates their diffusion and transport to the cells. A depository for hydrocarbon molecules is also formed, where the rate of hydrocarbon assimilation can be regulated to be optimal. The participation of lipids in promoting hydrocarbon assimilation applies to both the non-toxic *n*-hexadecane and the toxic *n*-hexane and diesel fuel. However, an increase in biofilm density, the formation of a more pronounced extracellular matrix, and the release of polysaccharides were only observed in the presence of *n*-hexane and diesel fuel. In this case, the relatively hydrophobic, lipid-rich matrix is ‘diluted’ by hydrophilic β-polysaccharides, resulting in the formation of a buffer zone with an optimal hydrophilic–lipophilic balance. The increased extracellular matrix acts as a thick barrier that prevents the rapid infusion of toxic hydrocarbons. Small molecules of *n*-hexane and small diesel fuel compound slowly move within the porous, viscous, multi-channel β-polysaccharide-saturated matrix. Such a matrix is neither required nor favorable in the case of *n*-hexadecane.

Lipids are thought to enable bacterial adhesion. As hydrophobic substances, these compounds reduce the total surface energy and facilitate the approach of cells to the support for further multi-point adhesion [49]. Unfortunately, the adhesion potential of EPS lipids could not be confirmed by direct measurements of adhesion forces using AFM. This method cannot distinguish between the EPS components (lipids or carbohydrates) that facilitate or prevent *Rhodococcus* cells adhesion to polystyrene. The adhesion forces were uniformly distributed along the EPS layer on the glass surface. Some highly adhesive spots with adhesion forces ranging from 22 to 116 nN were present on this layer, but their chemical composition was unknown (Figure 6 and Appendix A). Contact of the EPS-modified AFM cantilever with either the lipid or carbohydrate component could explain the bimodal distribution of F_a_ values (Figure 7). The adhesion force of the EPS was lower than that of the unmodified cover glass. This was consistent with the results of the adhesive activity tests towards polystyrene. The adhesion of rhodococci was independent of the production of EPS carbohydrates and lipids by the cells. Apparently, EPS are not involved in the adhesion process of the studied IEGM strains to both relatively hydrophilic (cover glass and silicon cantilever) and relatively hydrophobic (polystyrene) surfaces. Lipids and polysaccharides were apparently proportionally mixed with each other within the EPS. Consequently, lipids had no significant effect on the adhesion of *R. ruber* IEGM 231 cells, which produce carbohydrate-rich EPS (Table 1, Figure 1a). Other factors must therefore be responsible for the adhesion of *Rhodococcus* cells to solid surfaces, such as the roughness of interacting surfaces and the presence of specific cytoadhesive structures on the cell surface [42]. Additionally, the manner in which the EPS cover the cell surfaces and form adhesive chains, as observed with the EPS produced by *Rhodococcus* sp. RC291 in the study [36], remains unclear.

## 5. Conclusions

For the first time, a large number of *Rhodococcus* strains belonging to well-known and ecologically significant species (*R. erythropolis*, *R. fascians*, *R. globerulus*, *R. jostii*, *R. opacus*, *R. rhodochrous* and *R. ruber*) and to less common species (*R. aetherivorans*, *R. cerastii*, *R. cercidiphylli*, *R. corynebacterioides*, *R. pyridinovorans*, *R. qingshengii* and *R. wratislaviensis*), as well as several unidentified *Rhodococcus* sp. isolates, were screened for their production of major EPS biopolymers, including polysaccharides, lipids, proteins, and nucleic acids. Loosely cell-bound EPS were characterized. The presence of large amounts of lipids, prevailing over or similar to the carbohydrate content, and low levels of proteins and nucleic acids in this fraction were detected. This chemical profile was consistent across all the *Rhodococcus* strains studied. Previous studies have mainly focused on the total fraction of loosely (separated by centrifugation only) and tightly (separated after heating) bound EPS, and this study properly characterized the loose fraction of *Rhodococcus* EPS. The chemical composition is important for estimating the biotechnological potential of EPS. Notably, *R. qingshengii* QDR4-2 produced non-acidic exopolysaccharides saturated with mannose residues, enhancing the antioxidant properties of these EPS [19]. The large amounts of lipids indicate the potential applications of crude EPS produced by the IEGM strains in the fields of hydrocarbon emulsification, bioremediation, and the development of emollient additives.

Another important finding was the strain-specific nature of the chemical composition and production of EPS. This relates to the genetic diversity of *Rhodococcus* strains within species. Further detailed analysis of EPS produced by individual strains is required, as is the identification of genes that code for EPS biosynthesis enzymes and an array of strains-producers for specific purposes. Future work will focus on purifying *Rhodococcus* EPS, identifying specific monomers, and screening functional characteristics. The aim is also to optimize growth conditions to increase the EPS yield, scale up production, and develop biotechnological products composed of EPS produced by *Rhodococcus* strains from the IEGM Collection.

## Figures and Tables

**Figure 1 polymers-17-01912-f001:**
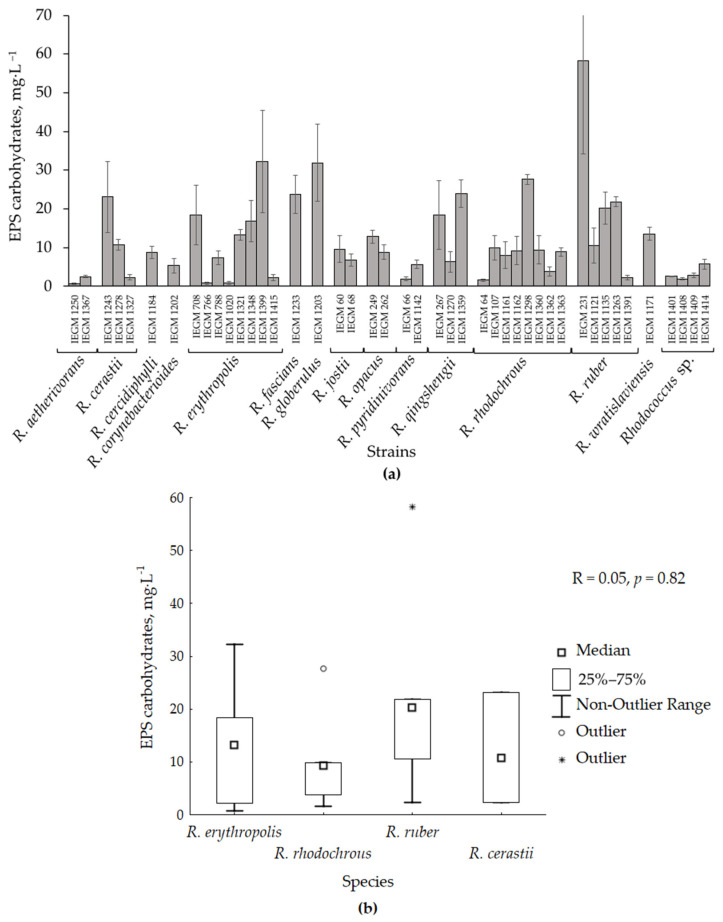
Production of EPS carbohydrates by various *Rhodococcus* species. Results are presented as: (**a**)—a diagram with strains grouped by species (means ± standard deviations are shown), (**b**)—box plots for EPS production by species (the total number of strains analyzed *n* = 26; the *R. erythropolis* box includes both *R. erythropolis* and *R. qinshengii* strains, https://lpsn.dsmz.de/species/rhodococcus-erythropolis, last accessed 17 June 2025). R—Spearman correlation coefficient; *p*—*p*-value.

**Figure 2 polymers-17-01912-f002:**
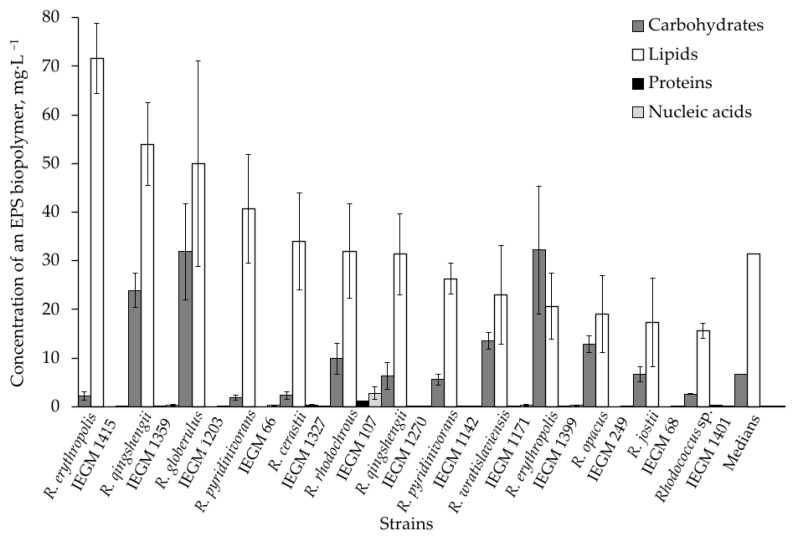
Comparison of major biopolymer concentrations in EPS produced by 13 *Rhodococcus* strains. Means ± standard deviations for individual strains are shown. The last set of columns shows the median biopolymer concentrations for all 13 strains.

**Figure 3 polymers-17-01912-f003:**
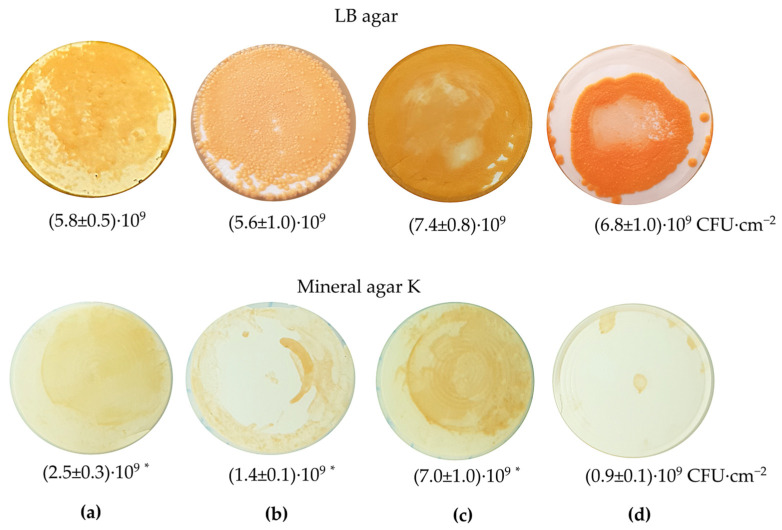
The *R. ruber* IEGM 231 biofilms grown on nitrocellulose filters placed on LB agar and mineral agar K in the presence of hydrocarbons. The hydrocarbons used: (**a**)—*n*-hexane, (**b**)—*n*-hexadecane, (**c**)—diesel fuel, (**d**)—control (no hydrocarbons). Biofilm densities are shown as means ± standard deviations, CFU·cm^−2^. * Statistically different from the control at *p* < 0.05.

**Figure 4 polymers-17-01912-f004:**
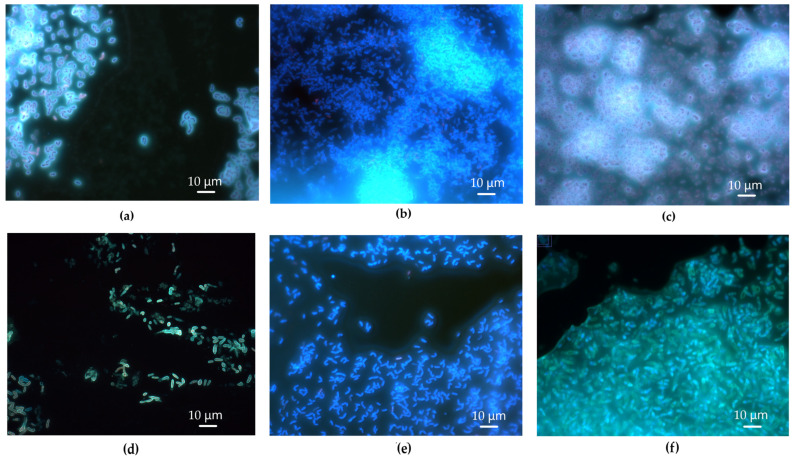
*R. ruber* IEGM 231 biofilms under a fluorescence microscope. Biofilms were obtained on: (**a**)—LB agar, (**b**)—LB agar in the presence of *n*-hexadecane, (**c**)—LB agar in the presence of diesel fuel, (**d**)—mineral agar K without externally added growth substrates, (**e**)—mineral agar K in the presence of *n*-hexadecane, and (**f**)—mineral agar K in the presence of *n*-hexane. Staining with Calcofluor White, which is specific to β-polysaccharides. Magnification ×1000.

**Figure 5 polymers-17-01912-f005:**
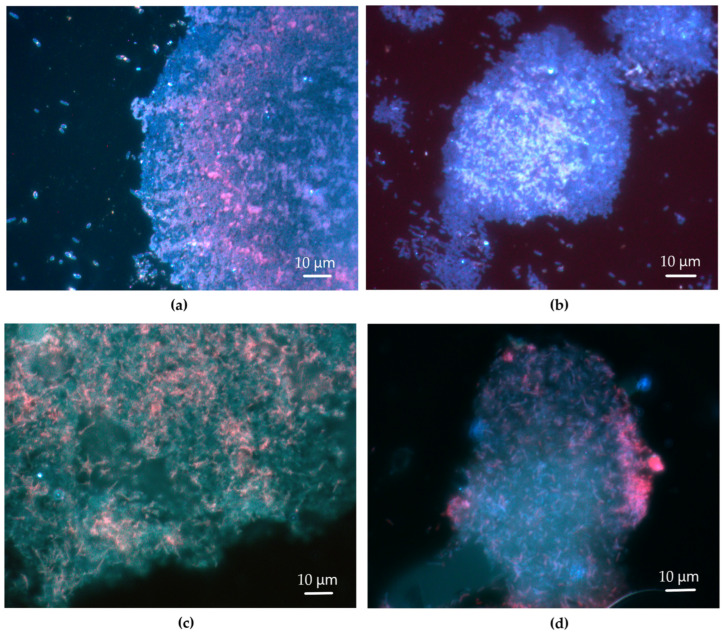
*Rhodococcus* cell aggregates under a fluorescence microscope. Strains: (**a**,**b**)—*R. aetherivorans* IEGM 1250, (**c**)—*R. erythropolis* IEGM 1020, (**d**)—*R. jostii* IEGM 68. Multi-component staining was performed using Calcofluor White Stain (specific to β-polysaccharides and exhibiting white-blue fluorescence), Wheat Germ Agglutinin Alexa Fluor^®^ 350 Conjugate (specific to glycoconjugates and exhibiting blue fluorescence) and FilmTracer SYPRO^®^ Ruby Biofilm Matrix Stain (specific to proteins and exhibiting red-orange, red or pink fluorescence). Magnification ×1000.

**Figure 6 polymers-17-01912-f006:**
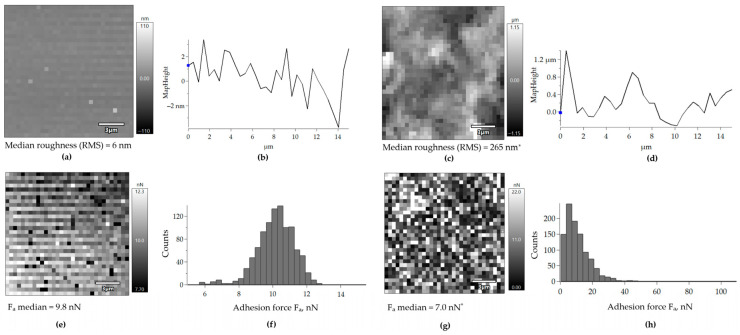
Distribution of adhesion forces F_a_ on the surface of unmodified cover glass (**a**,**b**,**e**,**f**) and of dried EPS extracted from the *R. ruber* IEGM 231 cells (**c**,**d**,**g**,**h**). The following data are shown: (**a**,**c**)—topographic maps, (**b**,**d**)—profiles of surfaces, (**e**,**g**)—adhesion force maps, and (**f**,**h**)—histograms showing the distribution of adhesion force values. RMS—root mean squared roughness. The slope of the topographic maps and surface profiles was eliminated using the Planefit function. * Significantly different from the control (unmodified cover glass) according to the Mann–Whitney U test at *p* < 0.05.

**Figure 7 polymers-17-01912-f007:**
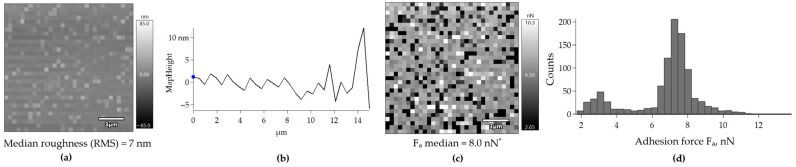
Distribution of adhesion forces F_a_ on the surface of an unmodified cover glass scanned with a cantilever modified with the *R. ruber* IEGM 231 EPS. The following data are shown: (**a**)—a topographic map, (**b**)—a profile of the scanned surface, (**c**)—an adhesion force map, and (**d**)—a histogram showing the distribution of adhesion force values. RMS—root mean squared roughness. The slope of the topographic maps and surface profiles was eliminated using the Planefit function. * Significantly different from the control (unmodified cantilever) according to the Mann–Whitney U test at *p* < 0.05.

**Figure 8 polymers-17-01912-f008:**
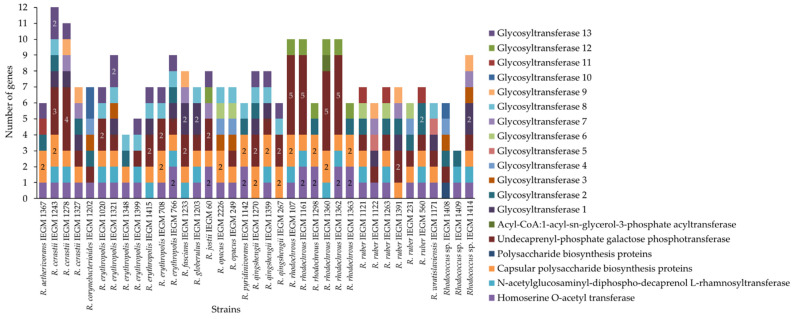
Combinations of genes that code for exopolysaccharide biosynthesis enzymes in *Rhodococcus* spp. If there is more than one copy of a gene in the genome, this is indicated by a number. IDs of glycosyltransferases (e.g., glycosyltransferase 1 and glycosyltransferase 2) correspond to the clades on the phylogenetic tree for genes (Figure 9).

**Figure 9 polymers-17-01912-f009:**
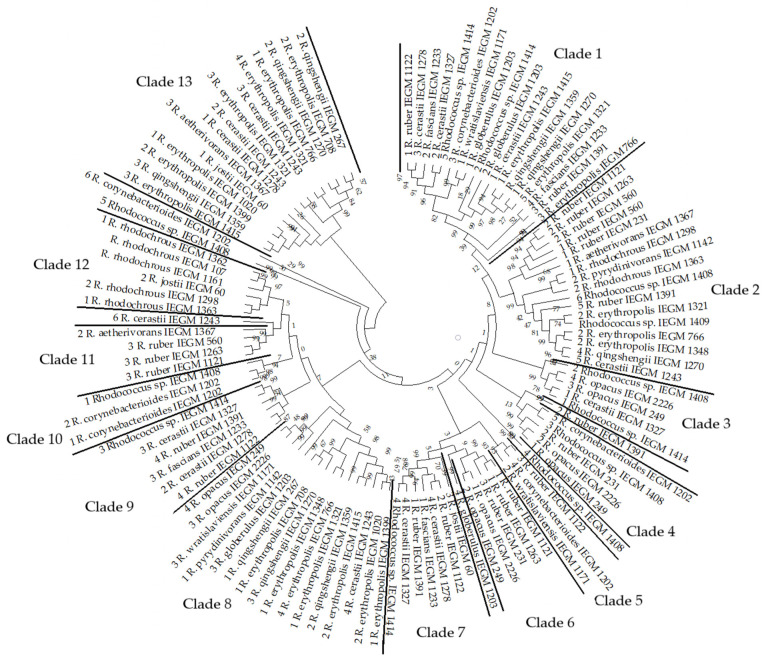
The phylogenetic tree of *Rhodococcus* glycosyltransferases, constructed using aligned nucleotide sequences. The tree was constructed using the neighbor-joining method of MEGA. The percentage of 500 bootstrap replicate support is shown for each node. The tree is rooted at the midpoint. Branch lengths reflect evolutionary distances (similarities) between genes.

**Figure 10 polymers-17-01912-f010:**
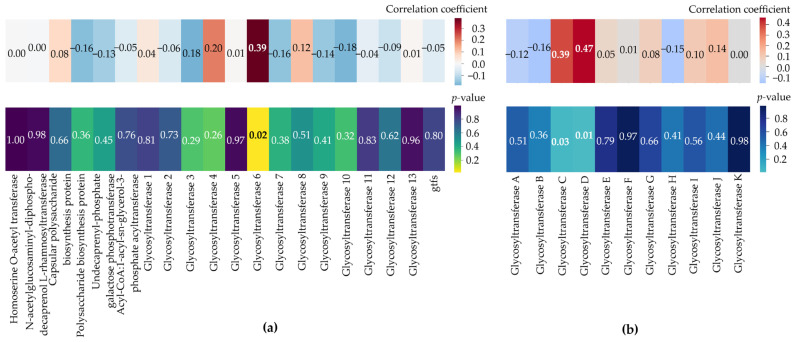
Correlation and *p*-value matrices (heatmaps) showing the dependence of EPS carbohydrate production by *Rhodococcus* cells on the presence and numbers of enzymes participating in exopolysaccharide biosynthesis. The glycosyltransferases are the same as those shown on the gene tree (Figure 9) (**a**) and on the protein tree (Appendix A) (**b**). Pearson correlation coefficients have been calculated and gtfs indicate the total number of glycosyltransferases in a given genome.

**Table 1 polymers-17-01912-t001:** Concentrations of the EPS carbohydrates produced by *Rhodococcus* spp.

Strain	EPS Carbohydrates, mg·L^−1^	Strain	EPS Carbohydrates, mg·L^−1^
*R. ruber* IEGM 231	58.2 ± 24.0	*R. cercidiphylli* IEGM 1184	8.8 ± 1.6
*R. erythropolis* IEGM 1399	32.2 ± 13.2	*R. opacus* IEGM 262	8.8 ± 1.9
*R. globerulus* IEGM 1203	31.9 ± 9.9	*R. rhodochrous* IEGM 1161	8.0 ± 3.5
*R. rhodochrous* IEGM 1298	27.6 ± 1.3	*R. erythropolis* IEGM 788	7.3 ± 1.8
*R. qingshengii* IEGM 1359	23.9 ± 3.5	*R. jostii* IEGM 68	6.7 ± 1.6
*R. fascians* IEGM 1233	23.7 ± 4.9	*R. qingshengii* IEGM 1270	6.3 ± 2.7
*R. cerastii* IEGM 1243	23.1 ± 9.2	*Rhodococcus* sp. IEGM 1414	5.7 ± 1.3
*R. ruber* IEGM 1263	21.8 ± 1.3	*R. pyridinivorans* IEGM 1142	5.6 ± 1.1
*R. ruber* IEGM 1135	20.2 ± 4.2	*R. corynebacterioides* IEGM 1202	5.3 ± 1.8
*R. erythropolis* IEGM 708	18.4 ± 7.7	*R. rhodochrous* IEGM 1362	3.8 ± 1.1
*R. qingshengii* IEGM 267	18.4 ± 8.8	*Rhodococcus* sp. IEGM 1409	2.9 ± 0.6
*R. erythropolis* IEGM 1348	16.8 ± 5.3	*Rhodococcus* sp. IEGM 1401	2.6 ± 0.1
*R. wratislaviensis* IEGM 1171	13.5 ± 1.7	*R. aetherivorans* IEGM 1367	2.5 ± 0.3
*R. erythropolis* IEGM 1321	13.2 ± 1.4	*R. cerastii* IEGM 1327	2.3 ± 0.7
*R. opacus* IEGM 249	12.8 ± 1.7	*R. ruber* IEGM 1391	2.3 ± 0.6
*R. cerastii* IEGM 1278	10.7 ± 1.3	*R. erythropolis* IEGM 1415	2.2 ± 0.8
*R. ruber* IEGM 1121	10.5 ± 4.6	*R. pyridinivorans* IEGM 66	1.9 ± 0.5
*R. rhodochrous* IEGM 107	9.9 ± 3.2	*Rhodococcus* sp. IEGM 1408	1.9 ± 0.3
*R. jostii* IEGM 60	9.6 ± 3.4	*R. rhodochrous* IEGM 64	1.6 ± 0.3
*R. rhodochrous* IEGM 1360	9.4 ± 3.7	*R. erythropolis* IEGM 1020	0.9 ± 0.4
*R. rhodochrous* IEGM 1162	9.2 ± 3.7	*R. erythropolis* IEGM 766	0.8 ± 0.3
*R. rhodochrous* IEGM 1363	8.9 ± 1.1	*R. aetherivorans* IEGM 1250	0.6 ± 0.2

Means ± standard deviations are shown.

**Table 2 polymers-17-01912-t002:** Concentrations of EPS biopolymers produced by *Rhodococcus* spp.

Strain	Lipids, mg·L^−1^	Proteins, mg·L^−1^	Nucleic Acids, mg·L^−1^
*R. erythropolis* IEGM 1415	71.7 ± 7.2	Below detectable level	0.167 ± 0.019
*R. qingshengii* IEGM 1359	54.0 ± 8.5	0.112 ± 0.014	0.363 ± 0.199
*R. globerulus* IEGM 1203	50.0 ± 21.1	Below detectable level	0.126 ± 0.067
*R. ruber* IEGM 1122	43.3 ± 12.2	Below detectable level	0.077 ± 0.019
*R. pyridinivorans* IEGM 66	40.7 ± 11.2	Below detectable level	0.252 ± 0.086
*R. cerastii* IEGM 1327	34.0 ± 9.9	0.365 ± 0.164	0.151 ± 0.027
*R. opacus* IEGM 2226	34.0 ± 1.4	0.301 ± 0.091	0.254 ± 0.024
*R. rhodochrous* IEGM 107	32.0 ± 9.7	1.131 ± 0.091	2.735 ± 1.276
*R. qingshengii* IEGM 1270	31.3 ± 8.4	0.014 ± 0.002	0.111 ± 0.014
*R. pyridinivorans* IEGM 1142	26.3 ± 3.2	0.023 ± 0.004	0.177 ± 0.004
*R. wratislaviensis* IEGM 1171	23.0 ± 10.1	0.094 ± 0.008	0.308 ± 0.131
*R. erythropolis* IEGM 1399	20.7 ± 6.8	0.043 ± 0.003	0.287 ± 0.079
*R. opacus* IEGM 249	19.0 ± 7.9	Below detectable level	0.078 ± 0.018
*R. jostii* IEGM 68	17.3 ± 9.1	Below detectable level	0.062 ± 0.024
*Rhodococcus* sp. IEGM 1401	15.6 ± 1.6	0.244 ± 0.032	0.181 ± 0.013

Means ± standard deviations are shown.

**Table 3 polymers-17-01912-t003:** Putative exopolysaccharide-related biosynthetic gene clusters in *Rhodococcus* genomes.

Most Similar Known Cluster	Similarity, %	Type	Number of Strains	Strains
ectoine Other	75	ectoine	1	*R. rhodochrous* IEGM 107
coelichelin NRP	27	NRPS	1	*R. erythropolis* IEGM 1321
oxalomycin B NRP + Polyketide	12	NRPS	1	*R. wratislaviensis* IEGM 1171
acarbose Saccharide	7	PKS-like, amglyccycl	5	*R. cerastii* IEGM 1243, *R. erythropolis* IEGM 788, IEGM 1020, IEGM 1321, *R. ruber* IEGM 1391
glycopeptidolipid Saccharide	7	NRPS	1	*R. opacus* IEGM 2226
maduramicin Polyketide + Saccharide	7	NRPS	1	*R. wratislaviensis* IEGM 1171
SF2575 Polyketide: Type II polyketide + Saccharide: Hydrid/tailoring saccharide	6	NRPS, terpene	24	*R. aetherivorans* IEGM 1367, *R. cerastii* IEGM 1243, IEGM 1278, IEGM 1327, *R. corynebacterioides* IEGM 1202, *R. erythropolis* IEGM 708, IEGM 766, IEGM 1020, IEGM 1321, IEGM 1348, *R. globerulus* IEGM 1203, *R. opacus* IEGM 249, IEGM 2226, *R. pyridinivorans* IEGM 1142, *R. rhodochrous* IEGM 107, IEGM 1360, IEGM 1362, *R. ruber* IEGM 231, IEGM 1121, IEGM 1391, *R. wratislaviensis* IEGM 1171, *Rhodococcus* sp. IEGM 1408, IEGM 1409, IEGM 1414
Coumermycin A1 Saccharide: Hydrid/tailoring saccharide + Other: Aminocoumarin	6	T1PKS	1	*R. ruber* IEGM 1391
hydromycin A Saccharide	6	arylpolyene	1	*R. cerastii* IEGM 1278
kendomycin B Polyketide	6	NRPS	1	*R. rhodochrous* IEGM 107
K-252a Alkaloid	5	NRPS-like	1	*R. ruber* IEGM 1391
prejadomycin/rabelomycin/gaudimycin C/gaudimycin D/UWM6/gaudimycin A Polyketide: Type II polyketide + Saccharide: Hydrid/tailoring saccharide	4	NRPS	1	*R. rhodochrous* IEGM 1362
Iomaiviticin A/Iomaiviticin C/Iomaiviticin D/Iomaiviticin E Polyketide: Type II polyketide + Saccharide: Hydrid/tailoring saccharide	3	NRPS	1	*R. qingshengii* IEGM 1359
EPS-related biosynthetic gene clusters are not found			16	*R. aetherivorans* IEGM 1250, *R. cercidiphylli* IEGM 1184, *R. erythropolis* IEGM 1399, IEGM 1415, *R. fascians* IEGM 1233, *R. jostii* IEGM 60, IEGM 68, *R. opacus* IEGM 262, *R. qingshengii* IEGM 1270, *R. rhodochrous* IEGM 1161, IEGM 1162, IEGM 1298, IEGM 1363, *R. ruber* IEGM 560, IEGM 1122, IEGM 1263

Abbreviations: amglyccycl—aminoglycoside/aminocyclitol biosynthesis, NRP—non-ribosomal peptide, NRPS—non-ribosomal peptide synthetase, and T1PKS—type I polyketide synthase.

## Data Availability

The raw data supporting the conclusions of this article will be made available by the authors upon request.

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
