# Peer review of "The Lipid- and Polysaccharide-Rich Extracellular Polymeric Substances of Rhodococcus Support Biofilm Formation and Protection from Toxic Hydrocarbons"

_polymers, 2025, doi:10.3390/polym17141912_

Round 1

Reviewer 1 Report

Comments and Suggestions for Authors

In this manuscript “The Lipid- and Polysaccharide-Rich Extracellular Polymeric Substances of Rhodococcus Support Biofilm Formation and Protection from Toxic Hydrocarbons ”, traditional studies have focused on the polysaccharide components in EPS, but this paper systematically reveals for the first time that lipids are the main components of Rhodococcus EPS (accounting for a much higher proportion than polysaccharides, proteins, and nucleic acids) , and that lipids are the main components of Rhodococcus EPS, the conventional view that“EPS is polysaccharide-centric” is challenged, providing new evidence for the chemical diversity of microbial EPS. The strain specificity (rather than species specificity) of rhodococcus EPS was clarified, reflecting the high genetic diversity of the genus bacteria and providing a reference for the study of microbial classification and functional evolution. The research is meaningful, but there are some problems with your manuscript. The comments and problems are as follows:

1.During EPS extraction, loosely bound EPS were isolated only by centrifugation, possibly underestimating the total EPS yield. Additional heating or chemical treatment steps are recommended to extract tightly bound EPS for a more comprehensive assessment of the total amount of EPS.

2.The data in tables 1 and 2 can be considered to be presented as histograms or heat maps to more intuitively compare the differences in EPS composition between different strains.

3.The overall language of the paper is fluent, but some sentences are longer. It is recommended to simplify to improve readability.

4.The Fluorescence microscope images in figures 4 and 5 are of low resolution. It is recommended that higher resolution images or partial magnification be provided to clearly show the EPS distribution.

5.Insufficient validation of gene function. It is suggested that the direct effect of specific glycosyltransferase on EPS synthesis could be verified by gene knockout or overexpression experiments.

  1. Biosynthetic gene cluster (BGCs) analysis found no clusters directly related to EPS, but other possible regulatory mechanisms were not discussed. Suggestions for additional discussions or experiments.

Author Response

Response to Reviewer #1

Reviewer #1

In this manuscript “The Lipid- and Polysaccharide-Rich Extracellular Polymeric Substances of Rhodococcus Support Biofilm Formation and Protection from Toxic Hydrocarbons ”, traditional studies have focused on the polysaccharide components in EPS, but this paper systematically reveals for the first time that lipids are the main components of Rhodococcus EPS (accounting for a much higher proportion than polysaccharides, proteins, and nucleic acids) , and that lipids are the main components of Rhodococcus EPS, the conventional view that“EPS is polysaccharide-centric” is challenged, providing new evidence for the chemical diversity of microbial EPS. The strain specificity (rather than species specificity) of rhodococcus EPS was clarified, reflecting the high genetic diversity of the genus bacteria and providing a reference for the study of microbial classification and functional evolution. The research is meaningful, but there are some problems with your manuscript. The comments and problems are as follows:

Authors

Thank you for a high estimation of our work, your kind attention to the study and your valuable comments. We have tried to answer all your questions and made all required corrections.

  1. Q: During EPS extraction, loosely bound EPS were isolated only by centrifugation, possibly underestimating the total EPS yield. Additional heating or chemical treatment steps are recommended to extract tightly bound EPS for a more comprehensive assessment of the total amount of EPS.

A: We completely agree that total amount of EPS include loosely (LB) and tightly (TB) bound EPS. We connect the obtained low production of EPS, and EPS polysaccharides specifically, in our study with underestimation of these substances because only centrifugation step was used to extract EPS. However, we continue to focus on chemical composition of this fraction and its possible relationships with other fractions because of two reasons. The first is biotechnological suitability. Steps of heating, ultrasonication or addition of reagents (EDTA, formamide, formaldehyde, NaOH, NaCl, etc.) require extra time and resources. We are balancing between an “easy” way to obtain EPS + estimation of properties of obtained loose EPS and the prospective of more “difficult” way but obtaining EPS saturated with polysaccharides and clearer properties. The second reason is investigation of EPS, which directly build the most outer layer of the cell surface and rule interactions of rhodococcal cells, for example, with liquid water-immiscible and solid nutrient substrates, with solid surfaces for adhesion, biofilm formation and colonization, and with other cells for aggregate formation and “communication”.

Moreover, there are diffuse and not clearly determined terms for loosely bound, tightly bound and total EPS. In many studies, researchers do not highlight what fraction they have extracted. There are works where only centrifugation has been used (where possible, regimes are specified): Urai et al. 2002 Actinomycetologica + Urai et al. 2004 Actinomycetologica (10,000 g for 10 min), Khan et al. 2013 Archives of Oral Biology http://dx.doi.org/10.1016/j.archoralbio.2013.09.011 (13,559 g for 30 min), Czemierska et al. 2016 Biochemical Engineering Journal http://dx.doi.org/10.1016/j.bej.2016.04.015 + SzczeÅ› et al. 2016 Journal of Solid State Chemistry http://dx.doi.org/10.1016/j.jssc.2016.07.014 (double centrifugation at 9,200 rpm for 30 min), Frølund et al. 1996 Water Research + Hu et al. 2020 Science of the Total Environment https://doi.org/10.1016/j.scitotenv.2020.138160 + Zheng et al. 2024 Water Research https://doi.org/10.1016/j.watres.2024.121331 (cation exchange resin was added, sample was stirred and then centrifugation at 12,000 g was conducted), Kim et al. 2024 Polymers https://doi.org/10.3390/polym16020244 (8,000 rpm for 15 min at 4 °C), and Santiso-Bellón et al. 2025 Gut Microbes https://doi.org/10.1080/19490976.2025.2469716. There are works where additional procedures have been used to separate both loose and tightly bound EPS (where possible, preliminary procedures are specified): Urai et al. 2007 Carbohydrate Research https://doi.org/10.1016/j.carres.2007.02.002 (CTAB), Maddela et al. 2018 Applied and Environmental Microbiology https://doi.org/10.1128/AEM.00756-18 + Madella Meng 2020 Science of the Total Environment https://doi.org/10.1016/j.scitotenv.2019.136402 (60 °C for 30 min), Li et al. 2023 Journal of Polymers and the Environment https://doi.org/10.1007/s10924-022-02604-0 (90 °C for 30 min), Feng et al. 2019 Chemical Engineering Journal https://doi.org/10.1016/j.cej.2019.122123 + Tang et al. 2023 Journal of Environmental Management https://doi.org/10.1016/j.jenvman.2023.119358 (1/168 37% formamide to separate LB-EPS with following addition of cation exchange resin to separate TB-EPS), Hu et al. 2020 International Journal of Biological Macromolecules https://doi.org/10.1016/j.ijbiomac.2019.12.228 (90 °C for 15 min), Tian et al. 2023 Environmental Pollution https://doi.org/10.1016/j.envpol.2023.121507 (80 °C for 40 min with following centrifugation at 8,000 rpm for 15 min to release LB-EPS and further treatment with 1.5 M NaOH and formaldehyde for 3 h with following centrifugation at 12,000 rpm for 30 min to release TB-EPS), and Weathers et al. 2015 Environmental Science and Technology https://doi.org/10.1021/es5060034 + Chen et al. 2025 Microorganisms https://doi.org/10.3390/microorganisms13010093 (heating, 20 g/L NaCl, EDTA).

It is a challenge to choose a standard extraction method. In summary, we attach to a position of Jiao et al. 2017 Applied Microbiology and Biotechnology http://doi.org/10.1007/s00253-017-8319-0. They separated LB-EPS and TB-EPS. They extracted LB-EPS, which they named mucoid EPS, by centrifugation at 12,000 g and 4 °C for 10 min and analyzed them. Further, they heated cell pellets at 70 °C for 3 h followed by centrifugation to extract TB-EPS and analyzed this fraction. We keep in mind a complex nature of total Rhodococcus EPS in terms of their composition and the force of binding to the cell wall for future research, but focus on a loose fraction in this specific study. Our position is framed in the manuscript (Introduction – p. 3, lines 120-134; Conclusions – p. 22, lines 766-783), as well as technical limitations of used analytical methods are discussed (Discussion – p. 19, lines 582-607 + pp. 19-20, lines 624-639).

  1. Q: The data in tables 1 and 2 can be considered to be presented as histograms or heat maps to more intuitively compare the differences in EPS composition between different strains.

A: EPS carbohydrate production by various strains has been additionally presented in the form of a diagram on Figure 1a, where carbohydrate concentrations were grouped by species. Consequently, Figure 1 has been modified. Modifications are highlighted in yellow. Data from Table 2 are also shown as a diagram in comparison with carbohydrate content on Figure 2. We would prefer to leave Tables in the manuscript in case extracting the Table values is needed, for example, for comparisons in future by other researchers. Moreover, production of EPS carbohydrates and EPS lipids is sorted from highest to lowest in Table 1 and Table 2, respectively, that, we hope, facilitate the search of strains with various level of EPS production. 

  1. Q: The overall language of the paper is fluent, but some sentences are longer. It is recommended to simplify to improve readability.

A: Several long sentences have been divided into shorter sentences. Separations are highlighted in yellow. The modifications are on p. 2 (lines 84-88), pp. 2-3 (lines 88-93), p. 7 (lines 310-312), p. 12 (lines 402-408), p. 13 (lines 461-464), p. 15 (lines 511-513), p. 17 (lines 562-564), p. 20 (lines 643-645), p. 22 (lines 771-774), and p. 24 (lines 839-842). Some sentences have been reduced. Repetitive sentences have been deleted. Sentences with lists (they are typically longer) have remained unmodified.

  1. Q: The Fluorescence microscope images in figures 4 and 5 are of low resolution. It is recommended that higher resolution images or partial magnification be provided to clearly show the EPS distribution.

A: Figures 4 and 5 have been saved with a higher resolution of 600 dpi. The figures are inserted into the manuscript and attached to the re-submission as separate .png files. Also, original images are attached to the submission, and each image can be opened as a separate .tiff file. Clearer focusing was a challenge as samples (biofilm fragments and aggregates) are volumetric.

  1. Q: Insufficient validation of gene function. It is suggested that the direct effect of specific glycosyltransferase on EPS synthesis could be verified by gene knockout or overexpression experiments.

A: This is right. Gene functions are only hypothesized in this study including revealed correlation between the EPS carbohydrate production and presence of gtf 6 in genomes. The current study includes only the bioinformatics analysis as a basement for further “wet” experiments with genes. Our first task was to estimate diversity of genes coded for EPS biosynthesis among the 47 Rhodococcus strains. A scaled bioinformatics analysis was performed towards search for genes coded for the synthesis of polysaccharides in the selected set of strains. A total of 25 genes in 37 strains (925 sequences) were analyzed and manually checked. We expected less diversity between strains of same species, and the revealed choppy species patterns were intriguing. Some genes in the certain proportion of strains were not automatically annotated (hypothetical proteins) and were found using only alignments. Future experiments with gene knockouts, estimation of the levels of expression and transcriptomics are required. We are afraid that involvement of these experiments would overweight this paper with data and discussion and would be more appropriate to constitute a separate paper. However, a bioinformatics part seems to be necessary for this manuscript as giving some basic explanations of differences in EPS biosynthesis by various Rhodococcus strains.  

  1. Q: Biosynthetic gene cluster (BGCs) analysis found no clusters directly related to EPS, but other possible regulatory mechanisms were not discussed. Suggestions for additional discussions or experiments.

A: We have analysed the surrounding areas of the genes of interest in Rhodococcus genomes. The discussion about putative roles of some transcriptional regulators found in these areas has been added (pp. 21-22, lines 713-737, highlighted in yellow).

The lack of association between putative polysaccharide biosynthetic genes and BGCs suggests that other regulatory mechanisms may govern exopolysaccharide biosynthesis in Rhodococcus. Analysing the surrounding area of the studied biosynthetic genes revealed presence of few genes coded for transcriptional regulators. These belong to the AraC, PhoU, XRE (xenobiotic response element) and OmpR families. The AraC (https://www.ncbi.nlm.nih.gov/Structure/sparcle/archview.html?archid=13263893, last accessed 8 July 2025), XRE (https://www.ebi.ac.uk/interpro/entry/cdd/CD02209/, last accessed 8 July 2025) and OmpR (https://www.uniprot.org/uniprotkb/P0AA18/entry, last accessed 8 July 2025) participate in the response of bacterial cells to stress, particularly the presence of xenobiotics in the case of XRE and osmotic stress in the case of OmpR. AraC is responsible for the metabolism of L-arabinose and the control of the degradation of several sugars. The PhoU regulates the metabolism of phosphorus [48], which, in the form of phosphates, is required for the transfer of glycosyl residues. We hypothesise that these transcriptional regulators can control the expression of (exo)polysaccharide biosynthetic genes in Rhodococcus. Additionally, their proximity to the studied genes, which mainly encode glycosyltransferases, capsular polysaccharide biosynthesis proteins and undecaprenyl-phosphate galactose phosphotransferases, suggest a role for EPS in protecting Rhodococcus cells from toxic hydrocarbons. For example, OmpR is part of a two-component sensor regulatory system. This system is responsible for the adaptation of bacteria to osmotic stress and reacts to the changes in the cell microenvironment. It can sense the disruptive effects on the cell wall and cytoplasmic membrane caused by toxicants such as n-hexane or diesel fuel. Further, it can impact the expression of genes involved in glycosylation and the transfer of glycosylated substances. This can result in the release of EPS components into the extracellular matrix, similar to the release of β-polysaccharides in the R. ruber IEGM 231 biofilms.

Reviewer 2 Report

Comments and Suggestions for Authors

The article is well written and has intriguing findings. However, a few queries need to be addressed before accepting the manuscript.  

  1. How does the composition of Rhodococcus EPS (especially the lipid-rich nature) influence their protective function against toxic hydrocarbons like n-hexane and diesel fuel?
  2. What evidence supports the claim that EPS production patterns in Rhodococcus are strain-specific rather than species-specific, and what might be the genetic basis for this variability
  3. What is the significance of the correlation between specific glycosyltransferase genes (e.g., clade 6 and clade D) and enhanced carbohydrate production in EPS?
  4. How did the presence of hydrocarbons affect the distribution of β-polysaccharides within the biofilm matrix, and what does this imply about Rhodococcus adaptation to toxic environments?
  5. What role did atomic force microscopy (AFM) reveal about the adhesive properties of EPS, and how might this adhesion variability influence biofilm stability on different surfaces?
  6. Why were no dedicated biosynthetic gene clusters (BGCs) for exopolysaccharide synthesis found in Rhodococcus genomes, and what alternative genetic elements might be responsible for EPS production?

Author Response

Response to Reviewer #2

Reviewer #2

The article is well written and has intriguing findings. However, a few queries need to be addressed before accepting the manuscript.

Authors

Thank you for a high estimation of our work, your kind attention to the study and your valuable comments. We have tried to answer all your questions and made all required corrections.

  1. Q: How does the composition of Rhodococcus EPS (especially the lipid-rich nature) influence their protective function against toxic hydrocarbons like n-hexane and diesel fuel?

A: It is typically considered that EPS act as buffer trapping ions and molecules of toxicants. Ions, especially cations, bind to negatively charged functional groups of exopolysaccharides (-OH and -COOH). Polar and charged organic molecules also can be hold by functional groups of exopolysaccharides. Concerning less polar or hydrophobic organic molecules, they can be trapped in pores and channels formed by exopolymers. This slows down transporting and diffusion of toxicants, and act as a barrier for their free moving. As thicker the barrier, as more pronounced its protective functions. Lipids can play a particular role in protection of rhodococcal cells from hydrocarbons. The studied hydrocarbons can be dissolved in the lipid-rich matrix. It can promote their assimilation and is true for all used hydrocarbons (n-hexane, n-hexadecane and diesel fuel). But for n-hexane and diesel fuel, which can rapidly reach the cell wall and cytoplasmic membrane due to the smaller molecule size or presence small size components (true for diesel, which is a mixture of compounds), formation a thick barrier saturated with polar / charged / hydrophilic beta-polysaccharides can be required. This matrix more efficiently regulates the rate of toxic hydrocarbon diffusion and can also act as a depository for hydrocarbon molecules regulating the rate of their assimilation. As a confirmation, we see increase in the biofilm density and release of beta-polysaccharides into the matrix in the presence of n-hexane and diesel fuel. But do not detect similar effects in the presence of non-toxic n-hexadecane, as adaptive reactions are not required in this case. Molecules – candidates to feel the disrupting effects of n-hexane and diesel fuel and stimulate growth of the biofilm matrix and release of polysaccharides can be assumed. These are transcriptional regulators found in proximity to biosynthetic genes. OmpR (tw-component response regulator) is among them. It is known for responsible for adaptation to osmotic stress but, as an environmental signal sensor, it reacts to microenvironment changes. As well as proteins can bind with ions and molecules via electrical, stereospecific and hydrophobic/hydrophilic interactions. But proteins are negligible in the studied fraction of loose EPS, which is the primary barrier at contact with toxic n-hexane and diesel fuel.

We have omitted to point the role of lipids in protection. As a correction, the discussion part has been modified, and additional explanations have been added (see p. 23, lines 792-807, highlighted in blue):

The EPS produced by Rhodococcus cells contains a high proportion of lipids. These lipids can play a significant role in the assimilation of hydrocarbons by rhodococcal cells. Lipids attract hydrocarbons, which are trapped and dissolved in the lipid-rich matrix. This probably facilitates their diffusion and transport to the cells. A depository for hydrocarbon molecules is also formed, where the rate of hydrocarbon assimilation can be regulated to be optimal. The participation of lipids in promoting hydrocarbon assimilation applies to both the non-toxic n-hexadecane and the toxic n-hexane and diesel fuel. However, an increase in a biofilm density, the formation of a more pronounced extracellular matrix, and the release of polysaccharides were only observed in the presence of n-hexane and diesel fuel. In this case, the relatively hydrophobic, lipid-rich matrix is ‘diluted’ by hydrophilic β-polysaccharides, resulting in the formation of a buffer zone with an optimal hydrophilic-lipophilic balance. The increased extracellular matrix acts as a thick barrier that prevents the rapid infusion of toxic hydrocarbons. Small molecules of n-hexane and small diesel fuel compound slowly move within the porous, viscous, multi-channel β-polysaccharide-saturated matrix. Such a matrix is neither required nor favorable in the case of n-hexadecane.

  1. Q: What evidence supports the claim that EPS production patterns in Rhodococcus are strain-specific rather than species-specific, and what might be the genetic basis for this variability

A: Statistical tests (Kruskal-Wallis and Spearman correlation) did not reveal statistically significant relationship between the concentration of carbohydrates in EPS and species. These results are shown in Figure 1b (Figure has been modified) and described in the text (p. 7, lines 295-301). It is seen also in Figure 1a, where strains were grouped by species, and strains of same species show various EPS carbohydrate production. The same is for lipid concentrations. Representatives of various species are placed in Table 2 and Figure 2 randomly, not arranged by species. Tendencies to have low amounts of carbohydrates, high amounts of lipids and almost have no nucleic acids and proteins in loose EPS are common for all Rhodococcus strains, the Rhodococcus genus in general. But variations in biopolymer concentrations in EPS are very high. The genetic basis can be highly diverse. The first idea is differences between the gene expression levels. However, it is seen from the literature (the references are cited in the paper) that monomer composition and proportions of various biopolymers in rhodococcal EPS are different. Probably, various biosynthetic pathways exist, which, we assume, are related with differences in genes. It was confirmed after the bioinformatics analysis. The studied Rhodococcus strains harbour from 3 to 12 different putative genes coded for the biosynthesis of only (exo)polysaccharides (Figure 8). Although gene combinations vary not very significantly between strains of same species, they vary that could be a sufficient reason for variations in EPS production levels. The revealed putative genes and their bioinformatics analysis, although more than 900 sequences have been analysed, are only a small part of the genetic basis for exopolymer production by rhodococci. Functions of genes are not confirmed in gene knockout and expression experiments, regulatory mechanisms are not studied, genes coded for EPS lipids are not analysed in this work, and transporters responsible for the release of polymers into the extracellular matrix are not studied. Strain specificity seems to be typical for many biological features of Rhodococcus actinomycetes. This is discussed in the work (see p. 20-21, lines 658-712).

  1. Q: What is the significance of the correlation between specific glycosyltransferase genes (e.g., clade 6 and clade D) and enhanced carbohydrate production in EPS?

A: These genes can be indicators and predictors of promising EPS producers. The corresponding sentence has been added to Discussion (p. 21, lines 690-691, highlighted in blue).

  1. Q: How did the presence of hydrocarbons affect the distribution of β-polysaccharides within the biofilm matrix, and what does this imply about Rhodococcus adaptation to toxic environments?

A: Possible mechanisms of how hydrocarbons affect the Rhodococcus biofilms are mentioned above in the answer for the first comment. Formation of thick extracellular matrix with a release of beta-polysaccharides from cells into the matrix occurs in the presence of n-hexane and diesel fuel as adaptation, not in the presence of n-hexadecane. Matrix becomes a homogeneously bright when exposed to these hydrocarbons. As seen from fluorescence images, no concentrated spots of polysaccharides within the matrix are formed. Diesel fuel probably is less toxic than n-hexane, and beta-polysaccharides form diffuse layers around cells in the presence of this hydrocarbon. It looks like an intermediate situation, a process of beta-polysaccharide releasing, which is completed when biofilms are exposed to more volatile n-hexane. AFM also did not reveal discrete spots or concentrates within the EPS. Adhesive points were randomly located on the surface of EPS drops on the cover glass.

  1. Q: What role did atomic force microscopy (AFM) reveal about the adhesive properties of EPS, and how might this adhesion variability influence biofilm stability on different surfaces?

A: AFM did not approve the role of EPS in bacterial cell adhesion in this study. Summarizing with adhesive activity tests (where no correlation between adhesion of rhodococcal cells to polystyrene and production of EPS was revealed), EPS apparently do not promote adhesion of rhodococci to both the hydrophilic (glass, silicon cantilever) and the relatively hydrophobic (polystyrene) surfaces. As shown in our previous work (Ivshina et al. Adhesion of Rhodococcus bacteria to solid hydrocarbons and enhanced biodegradation of these compounds. Sci. Rep. 2022, 12, 1–14, doi:10.1038/s41598-022-26173-3), other factors are responsible for the adhesion of Rhodococcus bacteria, such as roughness of interacting surfaces and cytoadhesive structures. But not EPS, apparently. Involvement of EPS in the adhesion of Rhodococcus has not been studied before by our team, and effects of this biopolymer mixture have been estimated in the current work. Discussion has been added to the text (p. 23, lines 808-831, highlighted in blue):

Lipids are thought to enable bacterial adhesion. As hydrophobic substances, these compounds reduce a total surface energy and facilitate the approach of cells to the support for further multi-point adhesion [49]. Unfortunately, the adhesion potential of EPS lipids could not be confirmed by direct measurements of adhesion forces using AFM. This method cannot distinguish between the EPS components (lipids or carbohydrates) that facilitate or prevent Rhodococcus cells adhesion to polystyrene. The adhesion forces were uniformly distributed along the EPS layer on the glass surface. Some highly adhesive spots with adhesion forces ranging from 22 to 116 nN were present on this layer, but their chemical composition was unknown (Figure 6, S5). Contact of the EPS-modified AFM cantilever with either the lipid or carbohydrate component could explain the bimodal distribution of Fa values (Figure 7). The adhesion force of the EPS was lower than that of the unmodified cover glass. This was consistent with the results of the adhesive activity tests towards polystyrene. The adhesion of rhodococci was independent of the production of EPS carbohydrates and lipids by the cells. Apparently, EPS are not involved in the adhesion process of the studied IEGM strains to both relatively hydrophilic (cover glass and silicon cantilever) and relatively hydrophobic (polystyrene) surfaces. Lipids and polysaccharides were apparently proportionally mixed with each other within the EPS. Consequently, lipids had no significant effect on the adhesion of R. ruber IEGM 231 cells, which produce carbohydrate-rich EPS (Table 1, Figure 1a). Other factors must therefore be responsible for the adhesion of Rhodococcus cells to solid surfaces, such as roughness of interacting surfaces and the presence of specific cytoadhesive structures on the cell surface [42]. Additionally, the manner in which the EPS cover the cell surfaces and form adhesive chains, as observed with the EPS produced by Rhodococcus sp. RC291 in the study [36], remains unclear.

  1. Q: Why were no dedicated biosynthetic gene clusters (BGCs) for exopolysaccharide synthesis found in Rhodococcus genomes, and what alternative genetic elements might be responsible for EPS production?

A: The areas surrounded biosynthetic genes were analysed. Most of genes in the proximity were hypothetical proteins. Annotated genes with functions in the proximity were connected with various metabolic processes (probably with oxidation, degradation, transport, resistance, metabolism of lipids / terpens / streoids / carbohydrates / amino acids / nucleotides). No mobile elements were detected in sectors with our genes of interest. And genes coded for transcriptional regulators were found. Role of latter is discussed on p. 21-22 (lines 713-737, highlighted in yellow). Text with description of surrounding genes has been added to Discussion (p. 22, lines 738-754, highlighted in blue):

Other genes in proximity to the putative genes coded for (exo)polysaccharide biosynthesis in Rhodococcus are connected with various processes. These genes encode transporters and enzymes that are likely to be involved in oxidation/degradation (oxidoreductases), resistance (e.g. chloramphenicol acetyltransferase) and the metabolism of lipids, terpens, steroids, carbohydrates, amino acids and nucleotides (various transferases, synthetases, dehydrogenases, dehydratases, etc.). Many genes are annotated as hypothetical proteins. This supports the hypothesis that EPS biosynthesis genes in Rhodococcus are involved in diverse metabolic processes and are not specific to the biosynthesis of (exo)polysaccharides. Genes coded for capsular polysaccharide biosynthesis proteins are always located next to any glycosyltransferase. Genes encoding undecaprenyl-phosphate galactose phosphotransferases are located in one sector with one or a few glycosyltransferases. These genes may be in direct proximity or separated by several other genes. We even hypothesise that these gtfs transfer galactose. The revealed gene distribution provides evidences for their coupling and involvement in one chain of metabolic reactions. Mobile elements were not detected in genome segments with polysaccharide biosynthesis genes. It appears that the diversity of rhodococcal gtfs is the result of evolutionary changes, e.g. divergence and convergence.

Round 2

Reviewer 2 Report

Comments and Suggestions for Authors

The authors have improved the manuscript and recommended it for publication.